# VARIATIONAL FEDERATED CONTINUAL LEARNING

## ABSTRACT

Federated continual learning (FCL) is an emerging learning paradigm with the potential to augment the scalability of federated learning by facilitating continual learning among multiple learners. However, FCL is beset by the significant challenges of local overfitting and catastrophic forgetting. To address both simultaneously, we propose Variational Federated Continual Learning (VFCL), a novel Bayesian neural network-based FCL framework that consists of two cores. First, we propose variational inference with mixture prior that merges global and local historical knowledge, which addresses local overfitting caused by the absence of global knowledge and catastrophic forgetting caused by the absence of historical knowledge simultaneously. Furthermore, to minimize the error in global knowledge acquisition, we present an effective global posterior aggregation method. Additionally, we provide a theoretical analysis on the upper bound of the generalization error of VFCL, which further helps to select the optimal hyperparameters. Empirical evaluations are conducted on VFCL, which outperforms other state-of-the-art methods on the widely used CIFAR100 and TinyImageNet datasets.

## 1 INTRODUCTION

Federated learning (FL) (McMahan et al., 2017; Bonawitz et al., 2019; Li et al., 2021; Kairouz et al., 2021) is a distributed learning paradigm that allows multiple learners to learn jointly while preserving data privacy. Its utility is demonstrated by its application in diverse domains such as finance, healthcare, and the Internet of Things. However, the typical FL paradigm is hindered by its static nature, which impedes the acquisition of new knowledge in dynamic environments, thereby seriously constraining the scalability of the model. Federated continual learning (FCL) (Yoon et al., 2021; Dong et al., 2022) was proposed to improve the ability of FL to maintain and expand knowledge among multiple learners in a collaborative manner. In FCL, each client performs continual learning (CL) on its own private task sequence while acquiring knowledge learned by other clients through FL. The data distribution of each client tends to be non-independent and identically distributed (non-i.i.d.) and changes over time. The client local training is prone to local overfitting (Zhang et al., 2022) because of the non-i.i.d. nature of the data distribution and limited data. Moreover, catastrophic forgetting (Kirkpatrick et al., 2017) during CL can also undermine model performance, leading to historical knowledge being forgotten. Addressing both local overfitting and catastrophic forgetting is a challenging problem.

Several recent studies have attempted to address these challenges in FCL. Regularization-based approaches (Yoon et al., 2021; Dong et al., 2022) exploit global and historical knowledge to mitigate catastrophic forgetting, but they neglect the client local overfitting problem. Knowledge distillation-based approaches (Ma et al., 2022; Usmanova et al., 2022) transfer knowledge from global and historical models to training model to retain learned knowledge, but they neglect local overfitting and lack explicit means of preventing overfitting. In addition, new network structures, such as prototypical networks (Hendryx et al., 2021), have been introduced to reduce communication and computational overheads. However, the neglect of local overfitting in existing work leads to the divergence of the client's local model from the global model, which reduces the generalization performance and convergence rate. Existing studies do not considered both local overfitting caused by the absence of global knowledge, and catastrophic forgetting caused by the absence of historical knowledge, which hinders them from addressing the challenges in FCL.

In this paper, we propose a novel Variational Federated Continual Learning (VFCL) framework based on Bayesian neural networks (BNNs) to address local overfitting and catastrophic forgetting

simultaneously. The VFCL framework consists of two cores, i.e., variational inference with mixture prior and global posterior aggregation. In variational inference with mixture prior, we merge global and local historical knowledge into a mixture prior that utilizes global knowledge to mitigate local overfitting while using local historical knowledge to address catastrophic forgetting. We represent global and local historical knowledge in the same knowledge space, balancing the two to better address local overfitting and catastrophic forgetting simultaneously. A BNN estimates the posterior distribution of parameters using only double the number of parameters, but learns an infinite ensemble, which makes the BNN more robust but also difficult to solve. Thus, we introduce variational inference to train the BNN efficiently. Second, to minimize the error in global knowledge acquisition, we introduce a novel aggregation method, which is proved to have minimal Shannon information loss compared to the original distributions. Additionally, We provide a theoretical analysis on the generalization error upper bound of VFCL, which can give the guidance for selecting optimal hyperparameters. The effectiveness of our VFCL is empirically evaluated on the CIFAR100 and TinyImageNet datasets. The evaluation results demonstrate that our VFCL outperforms other state-of-the-art methods. The main contributions of this study are summarized as follows:

- To our best knowledge, we are among the first to present a BNN-based FCL framework for addressing local overfitting and catastrophic forgetting simultaneously, which consists of a novel variational inference with mixture prior and an effective global posterior aggregation.
- Furthermore, we provide a theoretical analysis on the upper bound of the generalization error of our VFCL, which supports the design of our VFCL and the selection of hyperparameters.
- The extensive experiments are conducted by comparing VFCL with other methods on CIFAR100 and TinyImageNet and the results clearly demonstrate the superiority of our proposed VFCL.

## 2 RELATED WORK

### 2.1 FEDERATED CONTINUAL LEARNING

FCL (Yoon et al., 2021; Dong et al., 2022) is a distributed learning paradigm that allows multiple learners to collaborate while learning continuously on their own. However, it faces various challenges, such as privacy, communication efficiency, heterogeneity and local overfitting in FL (McMahan et al., 2017), and catastrophic forgetting in CL (Li & Hoiem, 2017). To address these issues, FedWeIT (Yoon et al., 2021) distinguishes between generic and task-specific parameters, uses parameter masks for selective knowledge migration, and applies regular terms to mitigate catastrophic forgetting. GLFC (Dong et al., 2022) uses weighted cross-entropy and distillation losses to solve class imbalances and catastrophic forgetting. In addition, many other regularization-based approaches(Zhang et al., 2023b; Luopan et al., 2023; Chaudhary et al., 2022; Wang et al., 2023; Qi et al., 2022) also attempt to mitigate catastrophic forgetting. CFeD (Ma et al., 2022) uses knowledge distillation on both the clients and server to transfer knowledge and mitigate catastrophic forgetting. Other knowledge distillation-based approaches (Usmanova et al., 2022; Zhang et al., 2023a) also focus on catastrophic forgetting. FedRecon (Hendryx et al., 2021) uses a prototypical network to transfer knowledge through category prototypes. However, existing studies have mainly focused on solving catastrophic forgetting while neglecting the relationship between global and local historical knowledge, which hinders the balanced treatment of local overfitting and catastrophic forgetting.

### 2.2 BAYESIAN NEURAL NETWORKS

In deep learning, Bayesian neural networks (MacKay, 1992) offer several advantages over traditional feedforward neural networks. Specifically, BNNs provide a means of regularization using the prior, and their structure naturally captures the uncertainty of weights (Blundell et al., 2015). However, performing exact Bayesian inferences on neural networks is infeasible owing to high computational overhead. In practice, the posterior is approximated using methods such as Markov Chain Monte Carlo (Gamerman & Lopes, 2006) and variational inference (Graves, 2011).

BNNs have been successfully applied to address local overfitting in FL. For example, FedPA (Al-Shedivat et al., 2020) employed a posterior estimation approach by averaging the local posterior distributions to obtain the global posterior distribution. FedEP (Guo et al., 2023) introduced expectation propagation in FL, obtained the global posterior through iteratively probabilistic message-passing between server and clients. In pFedGP (Achituve et al., 2021), a shared deep kernel function

is learned across clients, and each client trains a personalized Gaussian process classifier using a locally trained BNN. Similarly, in pFedBayes (Zhang et al., 2022), each client locally trains a BNN and uses the global posterior as a prior distribution to mitigate local overfitting.

Bayesian methods are also useful in CL, in which priors can be exploited to mitigate catastrophic forgetting. For example, VCL (Nguyen et al., 2017) partitions the posterior distribution across tasks into the posterior distribution of the old task and the likelihood of the current task, thereby enabling CL through recursive probability distribution computations. UCL (Ahn et al., 2019) evaluates the parameter uncertainty using a BNN and performs uncertainty-based adaptive parameter regularization. UCB (Ebrahimi et al., 2019), however, controls the learning rate of parameters based on the level of uncertainty captured by the BNN and constrains the variation of important parameters. Bui et al. (2018) proposed a partitioned variational inference framework that can be applied to FL and CL, but they do not consider the more challenging FCL problem. To the best of our knowledge, our proposed VFCL framework is the first BNN-based FCL approach.

## 3 METHODOLOGY

In this section, we first provide the problem definition in Sec. 3.1. Then, our proposed Variational Federated Continual Learning (VFCL) framework involves variational inference with mixture prior and global posterior aggregation, which are described in Secs. 3.2 and 3.3, respectively.

### 3.1 PROBLEM SETTING

FCL is a distributed learning paradigm with $C$ clients. A central server may exist to coordinate client training. Each client learns from its private task sequence $\mathcal{D}_c = \{\mathcal{D}_{c,t}\}_{t=1}^{T_c}$, where $T_c$ denotes the number of tasks of client $c$. An arbitrary task $\mathcal{D}_{c,t} = \{(\boldsymbol{x}_{c,t}^i, y_{c,t}^i)\}_{i=1}^{S_{c,t}}$, contains $S_{c,t}$ pairs of samples $\boldsymbol{x}_{c,t}$ and the corresponding labels $y_{c,t}$. Notably, the task sequences of different clients are independent, meaning that the number and content of their tasks can vary. To preserve client privacy, the raw data of $\mathcal{D}_{c,t}$ cannot be exchanged between clients. Considering the overhead of data storage and model retraining, the learner has access to data from the current task, and only a small amount of memory can be used to store the exemplars of the previous tasks.

The goal of FCL is to enable all clients to achieve better performance for their own personalized models by making use of the knowledge of other clients. In addition, clients should learn new tasks while maintaining their performance on the old tasks. The objective function can be written as

$$\min_{\{\boldsymbol{\theta}_c\}_{c=1}^{C}} \sum_{c=1}^{C} \mathcal{L}_c(\mathcal{D}_c; \boldsymbol{\theta}_c), \tag{1}$$

where $\mathcal{L}_c$ and $\boldsymbol{\theta}_c$ refer to the loss function and parameters of client $c$. As discussed above, the core challenge of FCL is how clients can better utilize global and historical knowledge for addressing issues of local overfitting and catastrophic forgetting. Therefore, as illustrated in Fig. 1, we propose a novel VFCL method involving variational inference with mixture prior and global posterior aggregation, whose details are given as follows.

### 3.2 VARIATIONAL INFERENCE WITH MIXTURE PRIOR

In VFCL, client $c$ trains a local BNN by variational inference to learn $\mathcal{D}_c$, which uses the approximate distribution $q(\boldsymbol{w}_c|\boldsymbol{\theta}_c)$ parameterized by $\boldsymbol{\theta}_c$ to estimate the posterior distribution $p(\boldsymbol{w}_c|\mathcal{D}_c)$ of the model weight $\boldsymbol{w}_c$. This can be described as the following optimization problem

$$\boldsymbol{\theta}_c^* = \arg\min_{\boldsymbol{\theta}_c} D_{\mathrm{KL}}[q(\boldsymbol{w}_c|\boldsymbol{\theta}_c)\|p(\boldsymbol{w}_c|\mathcal{D}_c))]. \tag{2}$$

Bayes by Backprop (Blundell et al., 2015) can be introduced to solve this problem, which assumes that $q(\boldsymbol{w}_c|\boldsymbol{\theta}_c)$ has a Gaussian probability density function with parameters $\boldsymbol{\theta}_c = (\boldsymbol{\mu}_c, \boldsymbol{\rho}_c)$, where $\boldsymbol{\mu}_c$ is the mean of the Gaussian distribution, and the standard deviation is parameterized as $\boldsymbol{\sigma}_c = \log(1 + \exp(\boldsymbol{\rho}_c))$ to ensure positivity. The model weight $\boldsymbol{w}_c$ can be sampled by $\boldsymbol{w}_c = \boldsymbol{\mu}_c + \boldsymbol{\sigma}_c \cdot \boldsymbol{\epsilon}$, where $\cdot$ denotes pointwise multiplication and $\boldsymbol{\epsilon}$ is a noise following standard normal distribution.

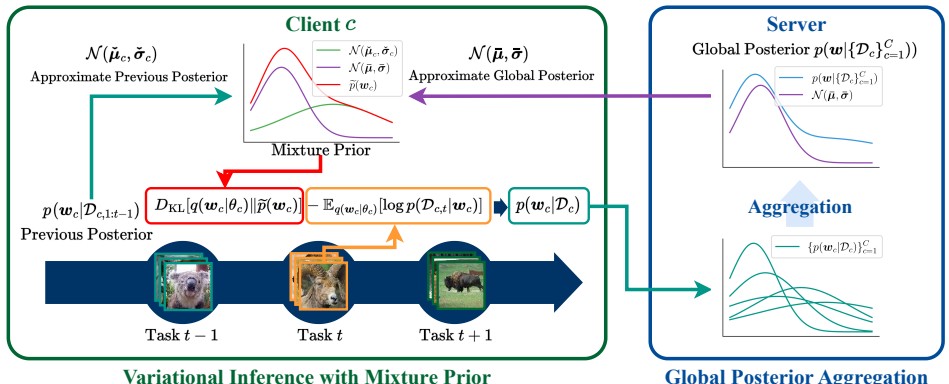

Figure 1: **Overview of VFCL.** The objective function of client $c$ includes the Kullback–Leibler divergence of $q(\boldsymbol{w}_c|\boldsymbol{\theta}_c)$ and the mixture prior $\widetilde{p}(\boldsymbol{w}_c)$ as well as the likelihood cost $-\mathbb{E}_{q(\boldsymbol{w}_c|\boldsymbol{\theta}_c)}[\log p(\mathcal{D}_{c,t}|\boldsymbol{w}_c)]$. Our main idea is to combine the previous posterior and the global posterior into a mixture prior $\widetilde{p}(\boldsymbol{w}_c)$ that solves both the catastrophic forgetting and overfitting encountered in FCL. In addition, we propose a global posterior aggregation method to obtain a more accurate $\mathcal{N}(\bar{\boldsymbol{\mu}}, \bar{\boldsymbol{\sigma}})$ from the most recent local posteriors $\{p(\boldsymbol{w}_c|\mathcal{D}_c)\}_{c=1}^C$.

The objective function is defined as

$$\mathcal{L}_c = \underbrace{D_{\mathrm{KL}}[q(\boldsymbol{w}_c|\boldsymbol{\theta}_c)\|p(\boldsymbol{w}_c)]}_{\text{KL-to-prior}} \underbrace{-\mathbb{E}_{q(\boldsymbol{w}_c|\boldsymbol{\theta}_c)}[\log p(\mathcal{D}_c|\boldsymbol{w}_c)]}_{\text{likelihood cost}}, \tag{3}$$

which consists of two parts: the Kullback-Leibler divergence between the approximate distribution $q(\boldsymbol{w}_c|\boldsymbol{\theta}_c)$ and the prior $p(\boldsymbol{w}_c)$, and the likelihood cost $-\mathbb{E}_{q(\boldsymbol{w}_c|\boldsymbol{\theta}_c)}[\log p(\mathcal{D}_c|\boldsymbol{w}_c)]$.

In FCL, client $c$ performs CL on its local private task sequence $\mathcal{D}_c$. The client's CL process can be expressed perfectly as a recursive posterior update process using the following equation

$$p(\boldsymbol{w}_c|\mathcal{D}_c) \propto p(\mathcal{D}_{c,t}|\boldsymbol{w}_c)p(\boldsymbol{w}_c|\mathcal{D}_{c,1:t-1}), \tag{4}$$

where the prior $p(\boldsymbol{w}_c|\mathcal{D}_{c,1:t-1})$ is the posterior distribution of the old task, and the likelihood $p(\mathcal{D}_{c,t}|\boldsymbol{w}_c)$ is for the task $t$ being learned. We consider $\mathcal{N}(\check{\boldsymbol{\mu}}_c, \check{\boldsymbol{\sigma}}_c)$ to be an approximate Gaussian distribution of $p(\boldsymbol{w}_c|\mathcal{D}_{c,1:t-1})$ with parameters $\check{\boldsymbol{\theta}}_c = (\check{\boldsymbol{\mu}}_c, \check{\boldsymbol{\rho}}_c)$.

Clients use global knowledge to overcome local overfitting. The global posterior can be considered as a finite mixture (McLachlan et al., 2019) of the local posteriors, which can be written as

$$p(\boldsymbol{w}|\{\mathcal{D}_c\}_{c=1}^C) = \sum_{c=1}^C \pi_c p(\boldsymbol{w}_c|\mathcal{D}_c), \tag{5}$$

where $\sum_{c=1}^C \pi_c = 1, \pi_c \geq 0$ and $p(\boldsymbol{w}_c|\mathcal{D}_c)$ is the most recent local posterior of client $c$. We consider $\mathcal{N}(\bar{\boldsymbol{\mu}}, \bar{\boldsymbol{\sigma}})$ to be an approximate Gaussian distribution of $p(\boldsymbol{w}|\{\mathcal{D}_c\}_{c=1}^C)$ with parameters $\bar{\boldsymbol{\theta}} = (\bar{\boldsymbol{\mu}}, \bar{\boldsymbol{\rho}})$.

To address catastrophic forgetting and local overfitting simultaneously, we combine the historical knowledge $\mathcal{N}(\check{\boldsymbol{\mu}}_c, \check{\boldsymbol{\sigma}}_c)$ and global knowledge $\mathcal{N}(\bar{\boldsymbol{\mu}}, \bar{\boldsymbol{\sigma}})$ to formulate the mixture prior as

$$\widetilde{p}(\boldsymbol{w}_c) = \prod_j^N \widetilde{p}(w_j) = \prod_j^N \left[ \lambda_k \cdot \mathcal{N}(w_j|\bar{\mu}_j, \bar{\sigma}_j) + (1-\lambda_k) \cdot \mathcal{N}(w_j|\check{\mu}_j, \check{\sigma}_j) \right], \tag{6}$$

where $\mathcal{N}(x|\mu, \sigma)$ is the Gaussian density evaluated at $x$ with mean $\mu$ and variance $\sigma^2$, $N$ denotes the number of parameters, and $\lambda_k$ is a hyperparameter. To conduct a more in-depth study of the mixture prior to FCL, we expand the mixture prior for a single parameter $w_j$ as

$$\widetilde{p}(w_j) = \frac{\lambda_k}{\sqrt{2\pi}\bar{\sigma}_j} \exp\left\{ -\frac{(w_j - \bar{\mu}_j)^2}{2\bar{\sigma}_j^2} \right\} + \frac{1-\lambda_k}{\sqrt{2\pi}\check{\sigma}_j} \exp\left\{ -\frac{(w_j - \check{\mu}_j)^2}{2\check{\sigma}_j^2} \right\}. \tag{7}$$

We note that $(w_j - \bar{\mu}_j)^2$ mitigates local overfitting by penalizing the inconsistency between the local and global models, and using the uncertainty $\bar{\sigma}_j$ as a weight. By contrast, $(w_j - \check{\mu}_j)^2$ mitigates

catastrophic forgetting by constraining the dissimilarity between the current and old models, and the importance of the parameter is measured by the uncertainty $\check{\sigma}_j$.

Taking advantage of $\widetilde{p}(\boldsymbol{w}_c)$, we rewrite Eq. 3 to obtain the final objective function as follows

$$\mathcal{L}_c = D_{\mathrm{KL}}[q(\boldsymbol{w}_c|\boldsymbol{\theta}_c)\|\widetilde{p}(\boldsymbol{w}_c)] - \mathbb{E}_{q(\boldsymbol{w}_c|\boldsymbol{\theta}_c)}[\log p(\mathcal{D}_{c,t}|\boldsymbol{w}_c)]. \tag{8}$$

As $\widetilde{p}(\boldsymbol{w}_c)$ is a complex mixture distribution, there is no closed-form solution to Eq. 8. Therefore, we approximate the solution using Monte Carlo sampling with the following objective function

$$\mathcal{L}_c \approx \sum_{k=1}^{n} \lambda_p \big( \log q(\boldsymbol{w}_{c,k}|\boldsymbol{\theta}_c) - \log \widetilde{p}(\boldsymbol{w}_{c,k}) \big) - \log p(\mathcal{D}_{c,t}|\boldsymbol{w}_{c,k}), \tag{9}$$

where $n$ is the number of samplings, and $\lambda_p$ is a hyperparameter that controls the effect of the prior.

To provide a clearer explanation, a graphical model of VFCL is presented in Fig. 2, which demonstrates the relationship between the priors and other components. The external plate denotes the repetitions of $C$ clients, whereas the internal plate denotes the repetitions of the number of samples for task $t$ for client $c$. The gray nodes indicate observed variables, whereas the white nodes indicate latent variables. The client local discriminant model is $p(y_{c,t}^i|\boldsymbol{x}_{c,t}^i, \boldsymbol{\theta}_c)$, whereas the client hidden variable $\boldsymbol{\theta}_c$ inherits the knowledge of global posterior $\bar{\boldsymbol{\theta}}$ and local previous posterior $\check{\boldsymbol{\theta}}_c$. In VFCL, global and local historical knowledge are aggregated into a mixture prior to balances their contributions.

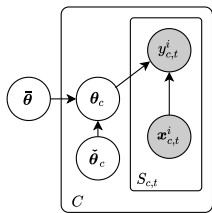

Figure 2: Graphical model of VFCL.

### 3.3 GLOBAL POSTERIOR AGGREGATION

The local posterior distributions of the clients are aggregated to obtain a more comprehensive global posterior distribution to mitigate local overfitting. A straightforward way to obtain the global posterior distribution is to minimize Kullback-Leibler divergence between the approximate global posterior $q(\boldsymbol{w}|\bar{\boldsymbol{\theta}})$ and local mixture posterior $q(\boldsymbol{w}_c|\boldsymbol{\theta}_c)$, which can be defined as

$$\bar{\boldsymbol{\theta}}^* = \arg\min_{\bar{\boldsymbol{\theta}}} D_{\mathrm{KL}}\big[q(\boldsymbol{w}|\bar{\boldsymbol{\theta}})\|\sum_{c=1}^{C} \pi_c q(\boldsymbol{w}_c|\boldsymbol{\theta}_c)\big]. \tag{10}$$

However, in FL, the data distributions are often non-i.i.d. across clients, which complicates the mixture distribution, and the solution of Eq. 10 becomes infeasible. Therefore, we introduce conflation (Hill, 2011) to fuse multiple distributions to a aggregated distribution, which is with density proportional to the product of the densities of input distribution. Conflation has the advantage of minimal information loss and high computational efficiency, with more details provided in Appendix E. We use conflation to obtain $q(\boldsymbol{w}|\bar{\boldsymbol{\theta}})$, and each parameter in $\bar{\boldsymbol{\theta}}$ is computed as follows

$$\bar{\mu}_j = \frac{\sum_{c=1}^{C} \frac{\mu_{c,j}}{\sigma_{c,j}^2}}{\sum_{c=1}^{C} \frac{1}{\sigma_{c,j}^2}}, \quad \bar{\sigma}_j^2 = \frac{1}{\sum_{c=1}^{C} \frac{1}{\sigma_{c,j}^2}}, \tag{11}$$

where $\mu_{c,j}, \sigma_{c,j}^2$ are the mean and variance of parameter $j$ on client $c$.

The differences between the proposed aggregation method and FedAvg (McMahan et al., 2017) are shown in Fig. 3. We consider $p(\boldsymbol{\theta}|\mathcal{D}_1)$ and $p(\boldsymbol{\theta}|\mathcal{D}_2)$ as the local posterior distributions of clients 1 and 2, respectively. The sum $\pi_1 p(\boldsymbol{\theta}|\mathcal{D}_1) + \pi_2 p(\boldsymbol{\theta}|\mathcal{D}_2)$ represents a mixture of the local posterior distributions of the clients, where $\pi_1 = \pi_2 = 0.5$. Ours and FedAvg denote the posterior distributions obtained using the two aggregation methods. The vertical dashed line denotes the highest probability in the distribution. By analyzing the shapes of the curves, we observe that our method tends to reduce the variance of the aggregated distribution, and the mean value of our method is closer to the location of highest

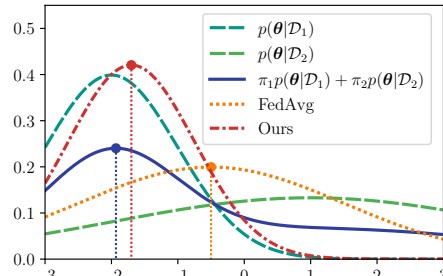

Figure 3: Comparison between our proposed aggregation method and FedAvg.

probability of $\pi_1 p(\boldsymbol{\theta}|\mathcal{D}_1) + \pi_2 p(\boldsymbol{\theta}|\mathcal{D}_2)$ than FedAvg. Our findings demonstrate that our method outperforms FedAvg and provides a more accurate approximation of the global posterior distribution.

# 4 THEORETICAL ANALYSIS

In this section, we provide a theoretical analysis on the upper bound of the generalization error of VFCL, which can further help to select the optimal hyperparameters. Particularly, we focus on the generalization error of the posterior distribution acquired through variational inference of any task on any client, therefore we omit the subscripts $c$ and $t$.

**Definition of generalization error.** Generalization error is defined in Eq. 12, where the distance between the likelihood $p(\mathcal{D}|\boldsymbol{w})$ and data distribution $p(\mathcal{D})$ is denoted by $d_H^2(,)$, and is used as an indicator of the error; $d_H^2(,)$ is the Hellinger distance defined in Eq. 13, as used in (Achituve et al., 2021; Bai et al., 2020). The optimal variational solution is $q(\boldsymbol{w}|\boldsymbol{\theta}^*)$, and $\boldsymbol{\Theta}$ represents the parameter space. The following equation represents the expectation or mean value of the generalization error

$$\int_{\boldsymbol{\Theta}} d_H^2\big(p(\mathcal{D}), p(\mathcal{D}|\boldsymbol{w})\big)q(\boldsymbol{w}|\boldsymbol{\theta}^*)d\boldsymbol{\theta}. \tag{12}$$

Hellinger distance measures the distance between two probability distributions, with a value such that $0 \leq d_H^2(,) \leq 1$. Notably, the Hellinger distance is a symmetric distance metric between two probability distributions.

$$d_H^2(P, Q) = \frac{1}{2}\|\sqrt{P} - \sqrt{Q}\|^2. \tag{13}$$

**Assumptions. 1.** The BNNs for all clients share the same architecture and contain $L$ hidden layers, all network layers are of width $M$, and the number of parameters is $N$. **2.** The activation function $\phi(\cdot)$ is 1-Lipschitz continuous. **3.** Constants $B$ exists such that $\|\boldsymbol{\theta}\|_\infty \leq B$.

**Remark.** We define the variational error $r$, approximation error $\xi$ and estimation error $\varepsilon$ as

$$r = \big(N(L+1)\log M + N\log(I\sqrt{S/N})\big)/S, \tag{14}$$

$$\xi = \inf_{\|\boldsymbol{\theta}\|_\infty \leq B} \|f_{\boldsymbol{\theta}} - f_0\|_\infty^2, \tag{15}$$

$$\varepsilon = \sqrt{r}\log S^\delta, \tag{16}$$

where $S$ is the number of samples contained in $\mathcal{D}$, $\delta > 1$, and $I$ is the dimension of the input data. We aim to approximate the ideal model $f_0$ using model $f_{\boldsymbol{\theta}}$, with $f_{\boldsymbol{\theta}^*}$ being the best approximation.

The optimization process for the local client is as follows

$$\boldsymbol{\theta}^* = \arg\inf_{q(\boldsymbol{w}|\boldsymbol{\theta})\in\mathcal{Q}} \big\{\lambda_p D_{\mathrm{KL}}[q(\boldsymbol{w}|\boldsymbol{\theta})\|\widetilde{p}(\boldsymbol{w})] + \int_{\boldsymbol{\Theta}} \mathcal{L}\big(p(\mathcal{D}), p(\mathcal{D}|\boldsymbol{w})\big)q(\boldsymbol{w}|\boldsymbol{\theta})d\boldsymbol{\theta}\big\}, \tag{17}$$

where $\mathcal{L}$ is defined as

$$\mathcal{L}\big(p(\mathcal{D}), p(\mathcal{D}|\boldsymbol{w})\big) = \log\frac{p(\mathcal{D})}{p(\mathcal{D}|\boldsymbol{w})}. \tag{18}$$

**Lemma 1.** Suppose that assumptions 1-2 are true, then the following inequality holds

$$\int_{\boldsymbol{\Theta}} d_H^2\big(p(\mathcal{D}), p(\mathcal{D}|\boldsymbol{w})\big)q(\boldsymbol{w}|\boldsymbol{\theta}^*)d\boldsymbol{\theta}$$
$$\leq \frac{1}{S}\big[\frac{1}{\lambda_p}\int_{\boldsymbol{\Theta}} \mathcal{L}\big(p(\mathcal{D}), p(\mathcal{D}|\boldsymbol{w})\big)q(\boldsymbol{w}|\boldsymbol{\theta}^*)d\boldsymbol{\theta} + D_{\mathrm{KL}}[q(\boldsymbol{w}|\boldsymbol{\theta}^*)\|\widetilde{p}(\boldsymbol{w})]\big] + C'\varepsilon^2, \tag{19}$$

where $C'$ is a positive constant.

**Lemma 2.** Suppose that assumptions 1-3 are true, then the following inequality holds

$$\int_{\boldsymbol{\Theta}} \mathcal{L}\big(p(\mathcal{D}), p(\mathcal{D}|\boldsymbol{w})\big)q(\boldsymbol{w}|\boldsymbol{\theta}^*)d\boldsymbol{\theta} + \lambda_p D_{\mathrm{KL}}[q(\boldsymbol{w}|\boldsymbol{\theta}^*)\|\widetilde{p}(\boldsymbol{w})] \leq S\big(C''\lambda_p r + C'''\xi\big), \tag{20}$$

where $C'', C'''$ are any diverging sequences.

**Theorem.** Combining Lemmas 1 and 2, we obtain the following upper bound for the generalization error

$$\int_{\boldsymbol{\Theta}} d_H^2\big(p(\mathcal{D}), p(\mathcal{D}|\boldsymbol{w})\big)q(\boldsymbol{w}|\boldsymbol{\theta}^*)d\boldsymbol{\theta} \leq C'\varepsilon^2 + C''r + \frac{C'''}{\lambda_p}\xi, \tag{21}$$

which is determined by the estimated error $\varepsilon$, variational error $r$, approximation error $\xi$, and hyperparameter $\lambda_p$. All of $C', C''$ and $C'''$ are constants. Here, $\varepsilon$ is affected by the sampling variability and stochastic nature of the data. $r$ is determined by variational approximation of the true posterior distribution, and $\xi$ measures the discrepancy between the approximate representation $f_{\boldsymbol{\theta}}$ and the ideal solution $f_{\boldsymbol{\theta}^*}$. Therefore, we make the following observations

- As the sample size $S$ increases, both $\varepsilon$ and $r$ decrease, and overall error decreases.
- As the model capacity (i.e., the number of parameters $N$) increases, $r$ increases and $\xi$ decreases.
- As the weight of the mixture prior $\lambda_p$ increases, $\xi/\lambda_p$ decreases, and overall error decreases.

Based on these observations, we can develop our method by choosing appropriate $N$ to balance $r$ and $\xi$. In addition, more importantly, we theoretically show that the mixture prior can reduce the upper bound of generalization error, which is beneficial to the model performance. Further, we also experimentally verified the effectiveness of the mixture prior in Tab. 4. Moreover, we found that a large $\lambda_p$, while reducing catastrophic forgetting, also reduces the model's adaptability to new tasks. Therefore, an appropriate $\lambda_p$ needs to be chosen to strike a balance between model stability and adaptability to new tasks.

## 5 EXPERIMENTS

### 5.1 IMPLEMENTATION DETAILS

VFCL was evaluated on two benchmark datasets: CIFAR100 (Krizhevsky et al., 2009) and TinyImageNet (Le & Yang, 2015). CIFAR100 comprises 100 categories with 60,000 images, and TinyImageNet contains 200 categories with 110,000 images. To set up the incremental task, we adopted the same strategy as in (Dong et al., 2022) by dividing the dataset into 10 tasks. Each task contained an equal number of categories, with no intersection between them. The number of clients for each task was determined randomly. To simulate a non-i.i.d. scenario, clients were instructed to select 50% of the classes in the current task, and the samples were then randomly selected from these classes. In addition, no overlap was observed in the samples between individual clients.

Comparative experiments were conducted using the following prominent methods in the FCL field: FedWeIT (Yoon et al., 2021), GLFC (Dong et al., 2022), CFeD (Ma et al., 2022), and FedRecon (Hendryx et al., 2021). In addition, we set up ablation experiments for the prior, including methods Ours w/o $\bar{\theta}$, Ours w/o $\check{\theta}$, Ours w/o $\bar{\theta} + \check{\theta}$, and a compared method named Ours-FedAvg that replaces global posterior aggregation with simple parameter averaging.

To ensure experimental fairness, models with the same number of parameters were used for the different methods. ResNet-34 (He et al., 2016) served as the backbone network for GLFC, CFeD and FedRecon, whereas FedWeIT and Our method used ResNet-18. In addition, the number of Monte Carlo samplings $n$ was set to 1 to ensure a fair computational overhead. All methods employ the same multi-head classifier as that in (Castro et al., 2018). Model training employed the SGD optimizer with a learning rate of 0.01 for all methods. The number of global iterations ($R$) for each task was set to 10, and the number of local iterations ($E$) was set to 5 for all methods. The exemplar management strategy followed (Rebuffi et al., 2017), in which 20 representative samples are stored for each category in the exemplar memory. The hyperparameters $\lambda_p$ and $\lambda_k$ were set to 1 and 0.5.

### 5.2 EXPERIMENTAL RESULTS

**Comparison experiments**. The results of the comparative experiments are presented in Tabs. 1 and 2, where the best results are in **bold** and the second-best results are underlined. All comparison experiments were run 3 times and averaged for final results. We observe that Ours outperformed the other methods (Yoon et al., 2021; Dong et al., 2022; Ma et al., 2022; Hendryx et al., 2021) by 1.6%–8.2% in terms of average accuracy, which validates the effectiveness of Ours. In addition, Ours performed best on all subsequent tasks beyond the first two warm-up tasks. Furthermore, the performance of Ours decreased more slowly than other methods, which indicates that Ours addresses the catastrophic forgetting more effectively.

**Comparison of convergence rates**. As shown in Fig. 4a, we compared the convergence rates of the different methods. FedRecon could not be compared because it is a non-iterative method. The figure shows the change in accuracy from Tasks 2 to 6, where each task contained 10 rounds of global communication. Ours exhibited a faster convergence rate in new task learning, and the upper bound reached by convergence was higher than those of the compared methods.

**Comparison of catastrophic forgetting**. As shown in Fig. 4b, we compared the catastrophic forgetting of the different methods. The accuracy rate gradually increases during the task-learning stage. However, the performance drops sharply once clients start learning the next task. Ours performs bet-

Table 1: Comparison of different methods in terms of accuracy (%) on CIFAR100 with 10 tasks.

| Task | 1 | 2 | 3 | 4 | 5 | 6 | 7 | 8 | 9 | 10 | Avg. | $\Delta$ |
|---|---|---|---|---|---|---|---|---|---|---|---|---|
| FedWeIT | 73.3 | 60.0 | 53.5 | 48.8 | 44.1 | 42.1 | 41.2 | 39.0 | 36.3 | 34.9 | 47.3 $_{\pm 1.7}$ | $\Downarrow$ 7.1 |
| GLFC | **77.6** | 64.2 | 56.6 | 52.1 | 47.0 | 45.7 | 45.8 | 43.9 | 40.5 | 39.9 | 51.3 $_{\pm 1.9}$ | $\Downarrow$ 3.1 |
| CFeD | 74.5 | **64.8** | 57.0 | 53.0 | 49.5 | 46.4 | 44.8 | 41.5 | 40.4 | 38.8 | 51.1 $_{\pm 1.8}$ | $\Downarrow$ 3.3 |
| FedRecon | 76.7 | 61.8 | 52.3 | 47.7 | 44.2 | 39.9 | 37.5 | 35.6 | 33.5 | 32.5 | 46.2 $_{\pm 1.8}$ | $\Downarrow$ 8.2 |
| Ours-FedAvg | 77.0 | 63.6 | 57.7 | 54.5 | 50.4 | 49.1 | 48.5 | 46.5 | 43.0 | 42.3 | 53.3 $_{\pm 1.1}$ | $\Downarrow$ 1.1 |
| Ours w/o $\bar{\theta} + \check{\theta}$ | 74.1 | 61.7 | 53.2 | 49.3 | 44.6 | 43.6 | 41.0 | 39.5 | 36.2 | 36.0 | 47.9 $_{\pm 0.7}$ | $\Downarrow$ 6.5 |
| Ours w/o $\bar{\theta}$ | 73.8 | 60.3 | 54.1 | 51.2 | 47.8 | 45.4 | 44.0 | 42.2 | 40.3 | 39.4 | 49.9 $_{\pm 0.9}$ | $\Downarrow$ 4.5 |
| Ours w/o $\check{\theta}$ | 73.5 | 57.4 | 49.6 | 46.9 | 43.0 | 41.2 | 39.5 | 37.9 | 35.7 | 34.7 | 46.0 $_{\pm 0.6}$ | $\Downarrow$ 8.4 |
| Ours | 76.3 | 64.5 | **60.0** | **55.8** | **52.2** | **50.0** | **49.9** | **47.5** | **44.3** | **43.8** | **54.4** $_{\pm 1.1}$ | - |

Table 2: Comparison of different methods in terms of accuracy (%) on TinyImageNet with 10 tasks.

| Task | 1 | 2 | 3 | 4 | 5 | 6 | 7 | 8 | 9 | 10 | Avg. | $\Delta$ |
|---|---|---|---|---|---|---|---|---|---|---|---|---|
| FedWeIT | 70.5 | 48.7 | 41.9 | 38.4 | 37.5 | 34.5 | 32.6 | 31.0 | 30.3 | 29.3 | 39.5 $_{\pm 1.1}$ | $\Downarrow$5.6 |
| GLFC | **76.0** | 51.7 | 46.8 | 41.2 | 36.2 | 37.9 | 33.6 | 32.1 | 25.6 | 26.6 | 40.8 $_{\pm 2.0}$ | $\Downarrow$4.3 |
| CFeD | 75.1 | **53.0** | 46.1 | 41.9 | 39.4 | 36.0 | 34.6 | 31.2 | 30.8 | 27.8 | 41.6 $_{\pm 0.9}$ | $\Downarrow$3.5 |
| FedRecon | 62.9 | 51.7 | 47.1 | 43.3 | 42.8 | 40.4 | 38.6 | 36.9 | 36.4 | 35.0 | 43.5 $_{\pm 2.0}$ | $\Downarrow$1.6 |
| Ours-FedAvg | 64.2 | 48.0 | 42.3 | 39.1 | 38.3 | 36.0 | 34.7 | 32.8 | 32.9 | 32.2 | 40.1 $_{\pm 0.5}$ | $\Downarrow$5.0 |
| Ours w/o $\bar{\theta} + \check{\theta}$ | 59.8 | 43.7 | 37.1 | 34.6 | 32.4 | 31.4 | 29.5 | 29.2 | 28.2 | 27.0 | 35.3 $_{\pm 1.1}$ | $\Downarrow$9.8 |
| Ours w/o $\bar{\theta}$ | 62.4 | 46.9 | 40.1 | 37.2 | 36.1 | 34.1 | 32.3 | 30.9 | 30.4 | 29.7 | 38.0 $_{\pm 1.3}$ | $\Downarrow$7.1 |
| Ours w/o $\check{\theta}$ | 62.2 | 46.3 | 39.9 | 37.0 | 35.4 | 34.0 | 32.4 | 30.5 | 30.2 | 29.4 | 37.7 $_{\pm 0.8}$ | $\Downarrow$7.4 |
| Ours | 63.9 | 51.6 | **47.7** | **45.6** | **44.6** | **41.5** | **42.0** | **39.0** | **37.9** | **37.5** | **45.1** $_{\pm 1.1}$ | - |

ter on the old task and remains stable as new tasks are added, indicating that Ours is more effective in addressing catastrophic forgetting.

**Comparison of global posterior aggregation**. We conducted comparative experiments between FedAvg and our aggregation method, the results are shown in Tabs. 1 and 2, where Ours-FedAvg indicates the aggregation using parameters averaging. The experimental results show that the performance of Ours-FedAvg was 1.1%–5.0% lower than Ours on CIFAR100 and TinyImageNet, indicating that our aggregation method is more suitable for the aggregation of posterior distributions.

**Comparison of communication and computation overheads**. We compared the models used by the different methods listed in Tab. 3, where the number of floating point operations (FLOPs) were obtained for inputs of size (1,3,224,224). Because both FedWeIT and Ours double the model parameters, ResNet-18 was used as the backbone network architecture, whereas GLFC, CFeD and FedRecon used ResNet-34. There are only minor differences in the number of parameters between the different models; therefore, the communication overheads are similar. However, the number of FLOPs for forward propagation in ResNet-34 is twice as high as ResNet-18 (Ours), which means that the clients require more computational resources.

Table 3: Comparison of different models in terms of computation complexity, including number of parameters (Params) and floating point operations (FLOPs).

| | ResNet-18 | ResNet-34 | ResNet-18 (FedWeIT) | ResNet-18 (Ours) |
|---|---|---|---|---|
| Params | 11.69M | 21.8M | 23.37M | 23.37M |
| FLOPs | 3.64G | 7.36G | 3.71G | 3.74G |

**Ablation studies on the mixture prior**. We investigated the contribution of the prior by conducting ablation studies. The results are listed in Tabs. 1 and 2, where Ours w/o $\bar{\theta}$ indicates that $\bar{\theta}$ is not considered as a prior, i.e., $\lambda_k = 0$, Ours w/o $\check{\theta}$ indicates that $\check{\theta}$ is not taken as a prior, i.e., $\lambda_k = 1$, and Ours w/o $\bar{\theta} + \check{\theta}$ indicates that the mixture prior is disabled, i.e., $\lambda_p = 0$. The results show that removing the prior causes a performance degradation of 4.5%–9.8%, which demonstrates that the prior has a positive effect on performance.

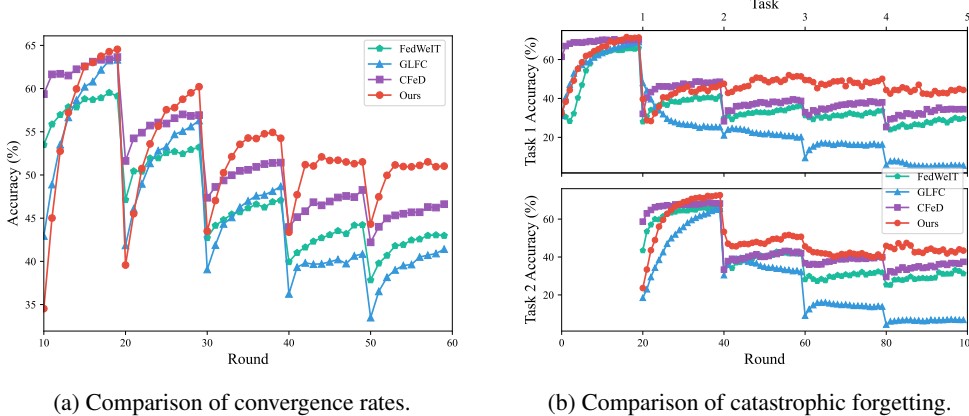

(a) Comparison of convergence rates.

(b) Comparison of catastrophic forgetting.

Figure 4: Comparison of convergence rates and forgetting rates on CIFAR100.

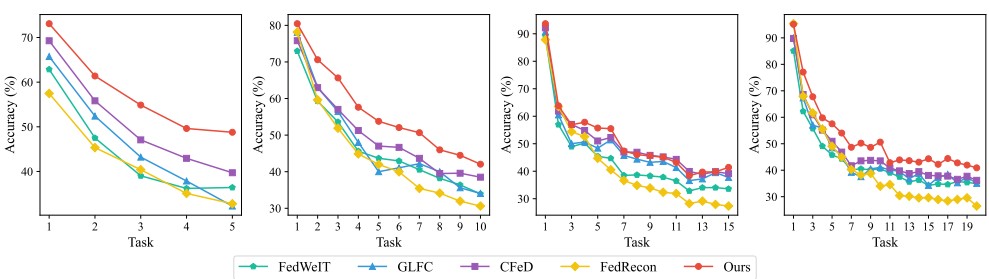

Figure 5: Results on CIFAR100 with different number of incremental tasks.

**Analytical experiments on the number of tasks**. We conducted comparative experiments by varying the number of tasks to further evaluate the efficacy of Ours, with results as presented in Fig. 5. The curves show that Ours achieved superior performance on average compared with the other methods. In addition, the slope of the Ours curve decreased less with an increasing number of tasks than was the case for the other methods, indicating that Ours exhibits less catastrophic forgetting of previously learned tasks.

**Analytical experiments on $\lambda_p$.** $\lambda_p$ controls the level of the prior constraint, and we investigated the performance changes under different values of $\lambda_p$. The results are shown in Tab. 4, where forgetting is calculated in the same manner as in (Yoon et al., 2021). A larger $\lambda_p$ causes the model to forget less about old knowledge, but at the same time, a large $\lambda_p$ also prevents the model from learning new knowledge. Thus, the setting of $\lambda_p$ must be a trade-off between adapting to new knowledge and avoiding forgetting old knowledge.

Table 4: Accuracy (%) and Forgetting (%) comparisons with different $\lambda_p$.

| $\lambda_p$ | Accuracy | Forgetting |
|---|---|---|
| 1e-4 | 47.1 | 43.6 |
| 1e-2 | 50.7 | 36.0 |
| 1 | 53.8 | 28.6 |
| 1e2 | 44.7 | 26.7 |
| 1e4 | 41.2 | 23.2 |

## 6 CONCLUSION

In this paper, we proposed a new Bayesian neural network-based FCL method named VFCL. Specifically, we propose variational inference with mixture prior to address overfitting and catastrophic forgetting. Then, a novel global posterior aggregation method is presented to reduce errors in posterior distribution aggregation. Additionally, we provide theoretical analysis and experimental comparisons on our VFCL method. Particularly, we show the upper bound of the generalization error of VFCL, and extensive experimental comparisons demonstrate the effectiveness of our proposed method for FCL task.

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

# A EXPERIMENTAL DETAILS

## A.1 DATASET

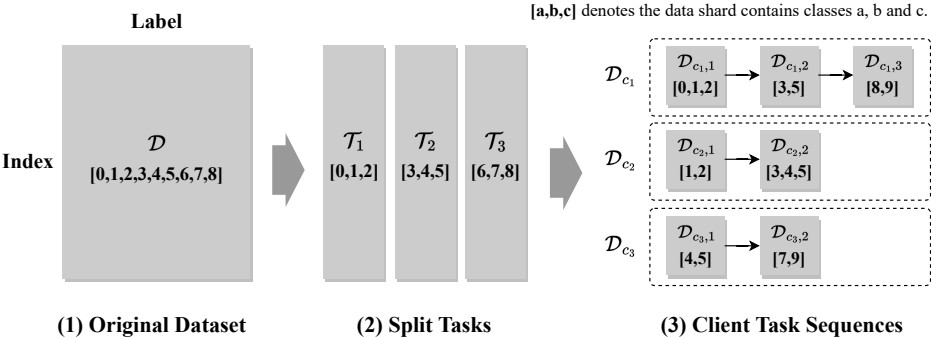

**(1) Original Dataset**  **(2) Split Tasks**  **(3) Client Task Sequences**

Figure 6: **Construction flow of client private task sequences.** First, the original dataset is split into several tasks that contain no overlapping classes. Second, the split tasks are randomly selected as the client's task and and the classes of tasks included are also randomly selected. Private task sequences from different clients may have similar (i.e., overlapping classes), unrelated, or interfering tasks.

The datasets CIFAR100 and TinyImageNet used in this paper are first split into a training set and a test set. Then the training set is partitioned in the manner of Fig. 6, and each client has a different sequence of tasks, their number of tasks, and the content of the tasks (i.e., the classes contained) are different. Such a dataset partitioning method is designed to simulate a real FCL scenario as much as possible, where each client learns on its own private sequence of tasks, and the local data distributions of the clients are non-i.i.d., the tasks between clients may be similar, unrelated, or interfering. Clients utilize the knowledge of other clients in the learning process to improve the performance of the local model while preventing interference from the knowledge of other clients. For the test set, each client is tested only on the classes it has learned, due to the fact that the tasks are personalized on each client.

## A.2 NETWORK ARCHITECTURE

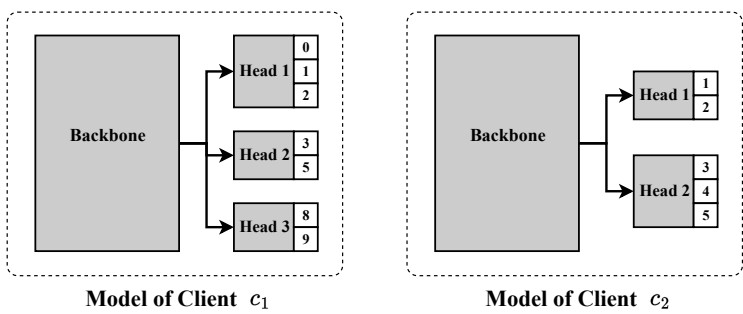

**Model of Client $c_1$**  **Model of Client $c_2$**

Figure 7: **Client model architecture.** The client's local model consists of two parts: the backbone network and the multi-head classifier. Backbone can adopt models like ResNet or DenseNet that have already been designed. The classifiers are dynamically constructed based on the tasks, where each head is a linear layer.

The architecture of the client local model is shown in Fig. 7. It is worth noting that the client model of VFCL is personalized, which is reflected in 2 aspects: (1) the weights of the backbone are personalized (2) the architecture and weights of the classifier are personalized. VFCL requires the backbone to be the same architecture across clients for knowledge exchange (i.e., global knowledge as a prior), however, the weights of the backbone are personalized to cope with the non-i.i.d. data of clients. In addition, since each client's task sequence is different, the multi-head classifier for

each client is also different, including the architecture and weights. The reasons for not sharing the knowledge of the classifier include: (1) it is heavily dependent on the client's local task, and is dynamically constructed based on the local task, and (2) the parameters of this part are very few and can be learned quickly at the client.

### A.3 METRICS

In the experiment, we followed (Yoon et al., 2021) to adopt accuracy and forgetting for evaluation. The accuracy after learning the $t$-th task is defined as follows:

$$A_t = \frac{1}{t} \sum_{i=1}^{t} a_{t,i}, \tag{22}$$

where $a_{t,i}$ denotes the accuracy of the $i$-th task after learning the $t$-th task. Incremental learning contains two evaluation metrics, task-IL and class-IL (Masana et al., 2022). Task-IL has access to the task-ID of the test sample during evaluation, and task-IL only needs to make predictions in the classes contained in the task to which the sample belongs. However, class-IL is unable to obtain the task-id, and needs to make predictions in all classes that have been learned. Generally, class-IL is more challenging than task-IL. In real FCL scenarios, the task-id of a sample can hardly be obtained, so we adopt class-IL to compute $a_{t,i}$.

The forgetting after learning the $t$-th task is defined as follows:

$$F_t = \frac{1}{t-1} \sum_{i=1}^{t-1} \max_{j \in 1, \ldots, t-1} (a_{j,i} - a_{t,i}), \tag{23}$$

which represents the average performance degradation of all old tasks.

## B OVERALL PIPELINE OF THE VFCL FRAMEWORK

The complete pipeline of VFCL comprises two parts: the server and client sides. On the server side, for each continual learning task $t$, the server first performs client selection and then performs $R$ global communications. In each global communication, the client trains the local BNN independently, and the server aggregates all updated BNN model parameters using Eq. 11. The client initializes with the latest global model and constructs $\bar{\theta}$ and $\check{\theta}_c$ as a mixture prior according to Eq. 6. The client then performs $E$ rounds of local iterations using stochastic gradient descent. The client first samples the parameter posterior distribution to obtain the model parameters in each batch, and calculates the loss according to Eq. 9, and then updates the parameters of the approximate posterior distribution $q(\boldsymbol{w}|\boldsymbol{\theta})$. In the last global iteration of each task, the client's local posterior distribution is retained as part of the prior when the new task is learned.

## C FURTHER EXPLANATION OF THE MIXTURE PRIOR

In this section, Eq. 7 is further explained, and the relationship between the mixture prior and pre-existing methods is discussed from the perspectives of FL and CL.

From the perspective of FL, the mixture prior and FedProx (Sahu et al., 2018) share the same idea, and both are used to alleviate local overfitting by penalizing the discrepancy between the local and global models. The difference lies in the coefficient $\bar{\sigma}_j$, which represents the uncertainty of the parameters. When the uncertainty of the parameter is low, it can be assumed that the parameter is deterministic in the global posterior distribution and has a high probability of being equally applicable locally; therefore, it is assigned a higher penalty weight. Conversely, parameters are considered uncertain in the global posterior distribution; therefore, smaller penalty weights are adopted.

From the perspective of CL, a similar idea was introduced in EWC (Kirkpatrick et al., 2017) and SI (Zenke et al., 2017) to mitigate catastrophic forgetting by protecting parameters that are important for previous tasks. The difference lies in the importance measure, for which EWC uses the Fisher information matrix, and SI uses the task-related information accumulated on the synapses. However, the mixture prior uses the uncertainty $\check{\sigma}_j$. Parameters with low uncertainty can vary only within a

---

**Algorithm 1** VFCL

---

**ServerExecutes($\{\boldsymbol{\theta}_c\}_{c=1}^C$):**
  define a synchronized task sequence containing $T$ tasks
  **for** task $t = 1, 2, ..., T$ **do**
    select clients with task $t$
    **for** round $r = 1, 2, .., R$ **do**
      **for** each client $c$ in parallel **do**
        $\boldsymbol{\theta}_c \leftarrow \text{ClientUpdate}(\bar{\boldsymbol{\theta}}, c, t, r)$
      $\bar{\boldsymbol{\theta}} \leftarrow$ Aggregation according to Eq. 11
  **return** $\{\boldsymbol{\theta}_c\}_{c=1}^C$

**ClientUpdate($\bar{\boldsymbol{\theta}}, c, t, r$):**
  construct the mixture prior according to Eq. 6
  **for** local epoch $e = 1, 2, ..., E$ **do**
    $\boldsymbol{\epsilon} \sim \mathcal{N}(0, \boldsymbol{I})$
    $\boldsymbol{\sigma} = \log(1 + \exp(\boldsymbol{\rho}))$
    $\boldsymbol{w} = \boldsymbol{\mu} + \boldsymbol{\sigma} \cdot \boldsymbol{\epsilon}$
    calculate loss $\mathcal{L}_c$ by Eq. 9
    $\boldsymbol{\theta}_c \leftarrow \boldsymbol{\theta}_c - \eta \nabla \mathcal{L}_c$
  **if** $r = R$ **then**
    $\breve{\boldsymbol{\theta}}_c \leftarrow \boldsymbol{\theta}_c$
  **return** $\boldsymbol{\theta}_c$

---

small range and are considered more important. Changes in these parameters can cause a sharp drop in the performance of previous tasks, therefore, their changes need to be penalized, and vice versa.

## D    PROOF OF LEMMAS

Detailed proof of the upper bound of the generalization error of VFCL is provided in this section.

### D.1    PROOF OF LEMMA 1

Before proving Lemma 1, we first introduce Eq. 24 from (Boucheron et al., 2013), where $q$ and $p$ are probability distributions and $h$ is any measurable function with $e^h \in L_1(q)$.

$$\log \int e^{h(\eta)} q(d\eta) = \sup_p \left[ \int h(\eta) p(d\eta) - D_{\text{KL}}[p \| q] \right]. \tag{24}$$

Next, we define $\eta\big(p(\mathcal{D}), p(\mathcal{D}|\boldsymbol{w})\big)$ as

$$\eta\big(p(\mathcal{D}), p(\mathcal{D}|\boldsymbol{w})\big) = \exp \left\{ \frac{1}{\lambda_p} \mathcal{L}\big(p(\mathcal{D}), p(\mathcal{D}|\boldsymbol{w})\big) + S d_H^2 \big(p(\mathcal{D}), p(\mathcal{D}|\boldsymbol{w})\big) \right\}. \tag{25}$$

We set $h(\eta) = \log \eta\big(p(\mathcal{D}), p(\mathcal{D}|\boldsymbol{w})\big)$, $q = \widetilde{p}(\boldsymbol{w})$, and $p = q(\boldsymbol{w}|\boldsymbol{\theta}^*)$, and Eq. 24 will be transformed into the following equation

$$\log \int_{\Theta} \eta\big(p(\mathcal{D}), p(\mathcal{D}|\boldsymbol{w})\big) \widetilde{p}(\boldsymbol{w}) d\boldsymbol{\theta} \geq$$
$$\int_{\Theta} \left[ \frac{1}{\lambda_p} \mathcal{L}\big(p(\mathcal{D}), p(\mathcal{D}|\boldsymbol{w})\big) + S d_H^2 \big(p(\mathcal{D}), p(\mathcal{D}|\boldsymbol{w})\big) \right] q(\boldsymbol{w}|\boldsymbol{\theta}^*) d\boldsymbol{\theta} - D_{\text{KL}}[q(\boldsymbol{w}|\boldsymbol{\theta}^*) \| \widetilde{p}(\boldsymbol{w})]. \tag{26}$$

Then, we obtain Eq. 27 from Eq. 26

$$\int_{\Theta} d_H^2 \big(p(\mathcal{D}), p(\mathcal{D}|\boldsymbol{w})\big) q(\boldsymbol{w}|\boldsymbol{\theta}^*) d\boldsymbol{\theta} \leq$$
$$\frac{1}{S} \Big[ \frac{1}{\lambda_p} \int_{\Theta} \mathcal{L}\big(p(\mathcal{D}), p(\mathcal{D}|\boldsymbol{w})\big) q(\boldsymbol{w}|\boldsymbol{\theta}^*) d\boldsymbol{\theta} \tag{27}$$
$$+ D_{\text{KL}}[q(\boldsymbol{w}|\boldsymbol{\theta}^*) \| \widetilde{p}(\boldsymbol{w})] + \log \int_{\Theta} \eta\big(p(\mathcal{D}), p(\mathcal{D}|\boldsymbol{w})\big) \widetilde{p}(\boldsymbol{w}) d\boldsymbol{\theta} \Big].$$

From Theorem 3.1 of (Pati et al., 2018), we obtain

$$\int_{\Theta} \eta\big(p(\mathcal{D}), p(\mathcal{D}|\boldsymbol{w})\big)\widetilde{p}(\boldsymbol{w})d\boldsymbol{\theta} \leq e^{C' S \varepsilon^2}. \tag{28}$$

By combining Eq. 28 and Eq. 27, we obtain Lemma 1 as

$$\int_{\Theta} d_H^2\big(p(\mathcal{D}), p(\mathcal{D}|\boldsymbol{w})\big)q(\boldsymbol{w}|\boldsymbol{\theta}^*)d\boldsymbol{\theta}$$
$$\leq \frac{1}{S}\Big[\frac{1}{\lambda_p}\int_{\Theta}\mathcal{L}\big(p(\mathcal{D}), p(\mathcal{D}|\boldsymbol{w})\big)q(\boldsymbol{w}|\boldsymbol{\theta}^*)d\boldsymbol{\theta} + D_{\mathrm{KL}}[q(\boldsymbol{w}|\boldsymbol{\theta}^*)\|\widetilde{p}(\boldsymbol{w})]\Big] + C'\varepsilon^2. \tag{29}$$

## D.2    PROOF OF LEMMA 2

The Kullback–Leibler divergence between two Gaussian distributions $\mathcal{N}(\mu_a, \sigma_a)$ and $\mathcal{N}(\mu_b, \sigma_b)$ is

$$D_{\mathrm{KL}}[\mathcal{N}(\mu_a, \sigma_a)\|\mathcal{N}(\mu_b, \sigma_b)] = \frac{1}{2}\Big[\log\Big(\frac{\sigma_b^2}{\sigma_a^2}\Big) + \frac{\sigma_a^2 + (\mu_a - \mu_b)^2}{\sigma_b^2} - 1\Big]. \tag{30}$$

Assuming that $\mu_b$ and $\sigma_b$ are variables, when Eq. 30 obtains the minimum value, the partial derivatives of Eq. 30 with respect to $\mu_b$ and $\sigma_b$ are 0, i.e.,

$$\frac{\partial D_{\mathrm{KL}}[\mathcal{N}(\mu_a, \sigma_a)\|\mathcal{N}(\mu_b, \sigma_b)]}{\partial \mu_b} = \frac{\mu_b - \mu_a}{\sigma_b^2} = 0, \tag{31}$$

$$\frac{\partial D_{\mathrm{KL}}[\mathcal{N}(\mu_a, \sigma_a)\|\mathcal{N}(\mu_b, \sigma_b)]}{\partial \sigma_b} = \frac{1}{\sigma_b} - \frac{\sigma_a^2 + (\mu_a - \mu_b)^2}{\sigma_b^3} = 0. \tag{32}$$

Eqs. 33 and 34 can be deduced from Eqs. 31 and 32 as

$$\mu_b = \mu_a, \tag{33}$$

$$\sigma_b^2 = \sigma_a^2 + (\mu_a - \mu_b)^2. \tag{34}$$

For any $k > 0$, the Kullback–Leibler divergence between any two mixture densities is bounded as

$$D_{\mathrm{KL}}\Big[\sum_{k=1}^K \pi_k g_k \Big\| \sum_{k=1}^K \widetilde{\pi}_k \widetilde{g}_k\Big] \leq \sum_{k=1}^K \pi_k \log\frac{\pi_k}{\widetilde{\pi}_k} + \sum_{k=1}^K \pi_k D_{\mathrm{KL}}[g_k\|\widetilde{g}_k]. \tag{35}$$

Next, we define $\widetilde{p}(\boldsymbol{w})$ as

$$\widetilde{p}(\boldsymbol{w}) = \prod_j^N \big[\lambda_k \cdot \mathcal{N}(\bar{\mu}_j, \bar{\sigma}_j^2) + (1 - \lambda_k) \cdot \mathcal{N}(\check{\mu}_j, \check{\sigma}_j^2)\big]. \tag{36}$$

For convenience, in the subsequent proof, we define $q(\boldsymbol{w}|\boldsymbol{\theta}^*)$ as

$$q(\boldsymbol{w}|\boldsymbol{\theta}^*) = \prod_j^N \big[\lambda_k \cdot \mathcal{N}(\mu_j^*, \sigma_j^{*2}) + (1 - \lambda_k) \cdot \mathcal{N}(\mu_j^*, \sigma_j^{*2})\big]. \tag{37}$$

To prove Lemma 2, we construct $\sigma^{*2}$ as Eq. 38, which is based on Assumption 3. Because Eq. 38 can be arbitrarily small, $\sigma^{*2} \leq B^2$ can easily be satisfied.

$$\sigma^{*2} = \frac{N}{8S(2BM)^{2(L+1)}\log 3IM\big[\big(s + 1 + \frac{1}{BM-1}\big)^2 + \frac{1}{(2BM)^2-1} + \frac{2}{(2BM-1)^2}\big]} \leq B^2. \tag{38}$$

With Eqs. 36 and 37 into Eq. 35, we obtain

$$
\begin{aligned}
D_{\mathrm{KL}}[q(\boldsymbol{w}|\boldsymbol{\theta}^*)\|\widetilde{p}(\boldsymbol{w})] &\leq \lambda_k \log\big(\frac{\lambda_k}{\lambda_k}\big) + (1-\lambda_k)\log\big(\frac{1-\lambda_k}{1-\lambda_k}\big) \\
&\quad + \lambda_k D_{\mathrm{KL}}[\mathcal{N}(\mu_j^*,\sigma_j^{*2})\|\mathcal{N}(\bar{\mu}_j,\bar{\sigma}_j^2)] + (1-\lambda_k)D_{\mathrm{KL}}[\mathcal{N}(\mu_j^*,\sigma_j^{*2})\|\mathcal{N}(\check{\mu}_j,\check{\sigma}_j^2)] \\
&= 0 + \frac{\lambda_k}{2}\sum_{j=1}^{N}\big[\log\big(\frac{\bar{\sigma}_j^2}{\sigma_j^{*2}}\big) + \frac{\sigma_j^{*2}+(\mu_j^*-\bar{\mu}_j)^2}{\bar{\sigma}_j^2} - 1\big] \\
&\quad + \frac{1-\lambda_k}{2}\sum_{j=1}^{N}\big[\log\big(\frac{\check{\sigma}_j^2}{\sigma_j^{*2}}\big) + \frac{\sigma_j^{*2}+(\mu_j^*-\check{\mu}_j)^2}{\check{\sigma}_j^2} - 1\big].
\end{aligned}
\tag{39}
$$

Combining Eqs. 33, 34, and 39, we can derive

$$
\frac{\sigma_j^{*2}+(\mu_j^*-\check{\mu}_j)^2}{\check{\sigma}_j^2} - 1 = 0,
\tag{40}
$$

$$
\check{\sigma}_j^2 = \sigma_j^{*2}+(\mu_j^*-\check{\mu}_j)^2 \leq \sigma_j^{*2}+B^2 \leq 2B^2,
\tag{41}
$$

$$
\frac{\sigma_j^{*2}+(\mu_j^*-\bar{\mu}_j)^2}{\bar{\sigma}_j^2} - 1 = 0,
\tag{42}
$$

$$
\bar{\sigma}_j^2 = \sigma_j^{*2}+(\mu_j^*-\bar{\mu}_j)^2 \leq \sigma_j^{*2}+B^2 \leq 2B^2.
\tag{43}
$$

Then we have

$$
D_{\mathrm{KL}}[q(\boldsymbol{w}|\boldsymbol{\theta}^*)\|\widetilde{p}(\boldsymbol{w})] \leq \big(\frac{\lambda_k}{2}+\frac{1-\lambda_k}{2}\big)N\log\frac{2B^2}{\sigma^{*2}} = \frac{N}{2}\log\frac{2B^2}{\sigma^{*2}}.
\tag{44}
$$

Combining Eq. 38 and Eq. 44 yields the following inequality

$$
\begin{aligned}
&D_{\mathrm{KL}}[q(\boldsymbol{w}|\boldsymbol{\theta}^*)\|\widetilde{p}(\boldsymbol{w})] \leq \\
&N(L+1)\log(2BM) + \frac{N}{2}\log\log(3IM) + N\log\big(4I\sqrt{\frac{S}{N}} + \frac{N}{2}\log(2B^2)\big).
\end{aligned}
\tag{45}
$$

Therefore, we obtain

$$
D_{\mathrm{KL}}[q(\boldsymbol{w}|\boldsymbol{\theta}^*)\|\widetilde{p}(\boldsymbol{w})] \leq C^{''}Sr.
\tag{46}
$$

For an arbitrary sample $(\boldsymbol{x},y)$, we can reformulate $\mathcal{L}\big(p(\mathcal{D}),p(\mathcal{D}|\boldsymbol{w})\big)$ as

$$
\begin{aligned}
\mathcal{L}\big(p(\mathcal{D}),p(\mathcal{D}|\boldsymbol{w})\big) &= \big(\|y-f_{\boldsymbol{\theta}}(\boldsymbol{x})\|_2^2 - \|y-f_0(\boldsymbol{x})\|_2^2\big)/2\sigma_\epsilon^2 \\
&= \big(\|y-f_0(\boldsymbol{x})+f_0(\boldsymbol{x})-f_{\boldsymbol{\theta}}(\boldsymbol{x})\|_2^2 - \|y-f_0(\boldsymbol{x})\|_2^2\big)/2\sigma_\epsilon^2 \\
&= \big(\|f_{\boldsymbol{\theta}}(\boldsymbol{x})-f_0(\boldsymbol{x})\|_2^2 + 2\langle y-f_0(\boldsymbol{x}), f_0(\boldsymbol{x})-f_{\boldsymbol{\theta}}(\boldsymbol{x})\rangle\big)/2\sigma_\epsilon^2.
\end{aligned}
\tag{47}
$$

Then we denote

$$
\mathcal{R}_1 = \int_{\boldsymbol{\Theta}} \|f_{\boldsymbol{\theta}}(\boldsymbol{x})-f_0(\boldsymbol{x})\|_2^2 q(\boldsymbol{w}|\boldsymbol{\theta}^*)d\boldsymbol{\theta},
\tag{48}
$$

$$
\mathcal{R}_2 = \int_{\boldsymbol{\Theta}} \langle y-f_0(\boldsymbol{x}), f_0(\boldsymbol{x})-f_{\boldsymbol{\theta}}(\boldsymbol{x})\rangle q(\boldsymbol{w}|\boldsymbol{\theta}^*)d\boldsymbol{\theta}.
\tag{49}
$$

For $\mathcal{R}_1$, since

$$
\|f_{\boldsymbol{\theta}}(\boldsymbol{x})-f_0(\boldsymbol{x})\|_2^2 \leq S\|f_{\boldsymbol{\theta}}-f_0\|_\infty^2 \leq S\big(\|f_{\boldsymbol{\theta}}-f_{\boldsymbol{\theta}^*}\|_\infty^2 + \|f_{\boldsymbol{\theta}^*}-f_0\|_\infty^2\big),
\tag{50}
$$

Then we can obtain

$$
\mathcal{R}_1 \leq S(r + \|f_{\boldsymbol{\theta}^*}-f_0\|_\infty^2).
\tag{51}
$$

For $\mathcal{R}_2$, since $y-f_0(\boldsymbol{x}) = \epsilon \sim \mathcal{N}(0,\sigma_\epsilon^2)$, then

$$
\mathcal{R}_2 = \epsilon^T \int_{\boldsymbol{\Theta}} \big(f_0(\boldsymbol{x})-f_{\boldsymbol{\theta}}(\boldsymbol{x})\big)q(\boldsymbol{w}|\boldsymbol{\theta}^*)d\boldsymbol{\theta} \sim \mathcal{N}(0, c_f\sigma_\epsilon^2),
\tag{52}
$$

where $c_f = \| \int_{\boldsymbol{\Theta}} \big( f_0(\boldsymbol{x}) - f_{\boldsymbol{\theta}}(\boldsymbol{x}) \big) q(\boldsymbol{w}|\boldsymbol{\theta}^*) d\boldsymbol{\theta} \|_2^2 \leq \mathcal{R}_1$ due to Cauchy-Schwarz inequality, then

$$\mathcal{R}_2 \leq C_R^{'} \mathcal{R}_1, \tag{53}$$

where $C_R^{'}$ is a positive constant or any diverging sequence.

By combining Eqs. 47, 51 and 53, we obtain

$$\begin{aligned}
\int_{\boldsymbol{\Theta}} \mathcal{L}\big(p(\mathcal{D}), p(\mathcal{D}|\boldsymbol{w})\big) q(\boldsymbol{w}|\boldsymbol{\theta}^*) d\boldsymbol{\theta} &= \mathcal{R}_1/2\sigma_\epsilon^2 + \mathcal{R}_2/\sigma_\epsilon^2 \\
&\leq (2C_R^{'} + 1)S(r + \| f_{\boldsymbol{\theta}^*} - f_0 \|_\infty^2)/2\sigma_\epsilon^2 \\
&\leq C_R^{''} S(r + \| f_{\boldsymbol{\theta}^*} - f_0 \|_\infty^2),
\end{aligned} \tag{54}$$

where $C_R^{''}$ is a positive constant or any diverging sequence.

Finally, we can obtain the following upper bound

$$\int_{\boldsymbol{\Theta}} \mathcal{L}\big(p(\mathcal{D}), p(\mathcal{D}|\boldsymbol{w})\big) q(\boldsymbol{w}|\boldsymbol{\theta}^*) d\boldsymbol{\theta} \leq C^{'''} S(r + \xi). \tag{55}$$

By combining Eq. 46 and Eq. 55, we obtain Lemma 2 as

$$\int_{\boldsymbol{\Theta}} \mathcal{L}\big(p(\mathcal{D}), p(\mathcal{D}|\boldsymbol{w})\big) q(\boldsymbol{w}|\boldsymbol{\theta}^*) d\boldsymbol{\theta} + \lambda_p D_{\mathrm{KL}}[q(\boldsymbol{w}|\boldsymbol{\theta}^*) \| \widetilde{p}(\boldsymbol{w})] \leq S(C^{''} \lambda_p r + C^{'''} \xi). \tag{56}$$

## E    DETAILED EXPLANATION OF GLOBAL POSTERIOR AGGREGATION

To simplify notation, we denote the local posterior distribution of client $c$ by $p_c$. The goal of global aggregation is to find a distribution $T(p_1, ..., p_C)$, which is able to integrate the posterior distributions of all clients. The most straightforward method for combining probability distributions is the weighted average, as follows

$$T(p_1, ..., p_C) = \sum_{c=1}^{C} \pi_c p_c, \tag{57}$$

where $\pi_c \geq 0$ and $\sum_{c=1}^{C} \pi_c = 1$. However, weighted average does not guarantee that the input and output distributions are of the same type, e.g., if $p_1, ..., p_C$ are unimodal Gaussian distributions and $T(p_1, ..., p_C)$ is generally a multimodal distribution. The distribution integrated by weighted average cannot be expressed in terms of the parameters of the input distribution, which prevents it from being applied to BNNs.

Another approach is to define a distribution $q_s$ of the same type as the $p_c$, by minimizing the following equation

$$D_{\mathrm{KL}}\big[q_s \| \sum_{c=1}^{C} \pi_c p_c \big], \tag{58}$$

we obtain an approximate distribution $q_s$ of $T(p_1, ..., p_C)$. However, $\sum_{c=1}^{C} \pi_c p_c$ is usually a complex multimode distribution, which makes it infeasible to solve Eq. 58. Although we can approximate the solution of Eq. 58 by Monte Carlo sampling, the huge computational overhead and the need for a training dataset make it infeasible in practice.

To address the problem of local posterior distributions aggregation across multiple clients, we introduce conflation (Hill, 2011). We denote the conflation of $p_1, ..., p_C$ by $Q(p_1, ..., p_C)$. According to Theorem 3.3 in (Hill, 2011), $Q(p_1, ..., p_C)$ is with density proportional to the product of the densities for each distribution, as follows

$$f(x) = \frac{\prod_{c=1}^{C} f_c(x) dx}{\int_{-\infty}^{\infty} \prod_{c=1}^{C} f_c(x) dy} \tag{59}$$

where $f_c$ is the probability density function of $p_c$. The distribution of each parameter in the BNN of our method is assumed to be Gaussian, so the aggregation of the parameter distributions is the product of the Gaussian distributions, as in Eq. 11.

Finally, we summarize the reasons why we use the conflation $Q = Q(p_1, ..., p_C)$ as follows:

- $Q$ of Gaussian distributions are always Gaussian.
- $Q$ has minimal Shannon information loss with the combined information from $p_1, ..., p_C$.
- $Q$ has a closed-form solution, which is computationally efficient.

# F ADDITIONAL EXPERIMENTAL RESULTS

## F.1 DIFFERENT BENCHMARK DATASETS

To further validate the effectiveness of VFCL, we conducted experiments on the two benchmarks proposed by (Yoon et al., 2021), including Overlapped-CIFAR100 and NonIID-50. For the fairness of the experiment, we follow the same experimental setting as in (Yoon et al., 2021), the backbone of FedWeIT is set to ResNet-18, and the backbone of other methods are set as in Tab. 3, The number of clients is set as 5, and each task has 20 rounds of global communication. The difference is that (Yoon et al., 2021) is evaluated with task-IL, while we adopt class-IL, which is more applicable to real scenarios and more challenging, as detailed in Sec. A.3. The results are shown in Tabs. 5 and 6, respectively. The best results are in **bold** and the second-best results are underlined. We observe that Ours outperformed the other methods (Yoon et al., 2021; Dong et al., 2022; Ma et al., 2022; Hendryx et al., 2021) by 1.3%–15.3% in terms of average accuracy.

Table 5: Comparison of different methods in terms of accuracy (%) on Overlapped-CIFAR100 with 10 tasks.

| Task | 1 | 2 | 3 | 4 | 5 | 6 | 7 | 8 | 9 | 10 | Avg. | $\Delta$ |
|---|---|---|---|---|---|---|---|---|---|---|---|---|
| FedWeIT | 68.4 | 50.6 | 42.0 | 38.4 | 34.0 | 33.7 | 32.8 | 31.6 | 28.2 | 31.8 | 39.1 $_{\pm 1.5}$ | $\Downarrow$9.1 |
| GLFC | 72.9 | 55.6 | 47.7 | 44.8 | 38.0 | 36.3 | 32.2 | 35.0 | 29.3 | 31.2 | 42.3 $_{\pm 1.2}$ | $\Downarrow$5.9 |
| CFeD | 69.3 | 57.4 | 52.1 | 46.0 | 43.0 | 40.4 | 36.4 | 35.6 | 33.7 | 33.6 | 44.8 $_{\pm 1.2}$ | $\Downarrow$3.4 |
| FedRecon | 71.9 | 59.2 | 40.3 | 34.8 | 31.8 | 31.8 | 28.6 | 27.9 | 31.3 | 30.7 | 38.8 $_{\pm 1.3}$ | $\Downarrow$9.4 |
| Ours | **74.7** | **60.0** | **53.0** | **50.0** | **45.2** | **42.9** | **41.0** | **40.9** | **37.4** | **37.3** | **48.2** $_{\pm 1.3}$ | - |

Table 6: Comparison of different methods in terms of accuracy (%) on NonIID-50 with 10 tasks.

| Task | 1 | 2 | 3 | 4 | 5 | 6 | 7 | 8 | 9 | 10 | Avg. | $\Delta$ |
|---|---|---|---|---|---|---|---|---|---|---|---|---|
| FedWeIT | 71.0 | **59.3** | 54.5 | **48.9** | **41.3** | 43.7 | 37.6 | 41.9 | 39.8 | 41.0 | 47.9 $_{\pm 0.6}$ | $\Downarrow$1.3 |
| GLFC | 69.7 | 41.7 | 49.7 | 35.5 | 26.4 | 30.6 | 33.1 | 35.0 | 36.6 | 34.5 | 39.3 $_{\pm 1.2}$ | $\Downarrow$9.9 |
| CFeD | **74.7** | 39.6 | 37.5 | 37.4 | 35.1 | 34.8 | 33.9 | 35.9 | 36.0 | 36.8 | 40.2 $_{\pm 1.2}$ | $\Downarrow$9.0 |
| FedRecon | 62.6 | 40.8 | 38.7 | 29.4 | 25.8 | 27.9 | 26.2 | 29.6 | 28.9 | 29.4 | 33.9 $_{\pm 1.5}$ | $\Downarrow$15.3 |
| Ours | 69.6 | 51.6 | **59.3** | 46.4 | 38.8 | **45.0** | **42.8** | **46.5** | **46.0** | **46.1** | **49.2** $_{\pm 0.7}$ | - |

## F.2 ANALYTICAL EXPERIMENTS ON $\lambda_k$

$\lambda_k$ is designed to control the balance between historical and global knowledge, as shown in Eq. 6. We have conducted analytical experiments on $\lambda_k$ to compare the accuracy and forgetting rate of different $\lambda_k$. The results are shown in Tab. 7. According to the results, we found that both historical and global knowledge play important roles in the model performance, and the lack of either is detrimental to the model performance. In addition, we found that as $\lambda_k$ increases, $1 - \lambda_k$ decreases, i.e., the weight of historical knowledge decreases, which causes an increase in forgetting.

Table 7: Accuracy and forgetting comparisons with different $\lambda_k$.

| $\lambda_k$ | $1 - \lambda_k$ | Accuracy (%) | Forgetting (%) |
|---|---|---|---|
| 0 | 1 | 49.3 | 32.2 |
| 0.25 | 0.75 | 51.4 | 34.7 |
| 0.5 | 0.5 | 53.8 | 28.6 |
| 0.75 | 0.25 | 51.6 | 35.4 |
| 1 | 0 | 45.5 | 40.5 |

### F.3  Analytical Experiments on $n$

$n$ in Eq. 9 denotes the number of Monte Carlo samplings, and we set $n = 1$ in the comparison experiments to achieve fairness in terms of computational overhead. In practice, it is a worthwhile trade-off for better model performance by adding some computational overhead. Therefore, we conducted further analytical experiments for different $n$. We compared the model accuracy and training time for $n$ from 1 to 10, and the results are shown in Fig. 8. It can be seen that when $n$ is between 1 and 4, the accuracy rises as $n$ increases, whereas when $n$ is greater than 4, larger $n$ does not provide more performance gains. The training time overhead is linearly correlated with $n$, and larger $n$ results in more computational overhead. Based on the results of the analytical experiments, we recommend setting $n$ to a small number (less than 4).

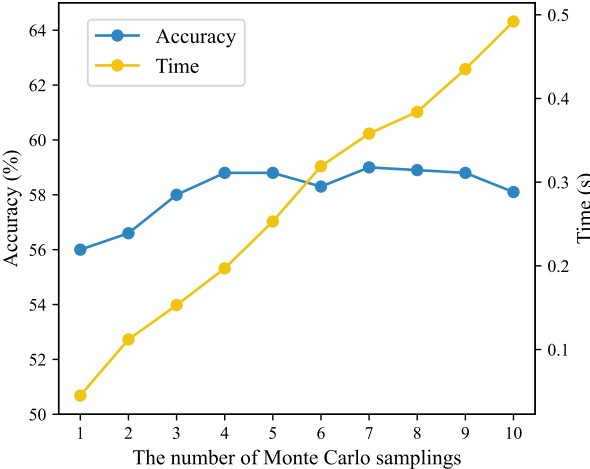

Figure 8: Comparison of different number of Monte Carlo samplings, where the time cost is the training time of a single batch which size is (32,3,224,224).

### F.4  Comparison of Computational Efficiency

We compared the time cost of training and testing of different models, as shown in Tab. 8. It can be seen that when $n = 1$, the training and testing time of ResNet-18 (Ours) is similar to that of ResNet-34 and ResNet-18 (FedWeIT), which shows that there is no drawback of our method in terms of time cost. When $n$ increases, the training time of ResNet-18 (Ours) increases linearly with $n$, while the testing time remains the same. This is because we use the mean of the learned parameter distributions as the sampling parameter in the test, and do not perform Monte Carlo sampling.

Table 8: Comparison of different models in terms of training and testing time, where the time is obtained by testing a single batch with size (32,3,224,224).

| Method | Setting | Training (s) | Testing (s) |
|---|---|---|---|
| ResNet-18 | - | $0.029_{\pm0.001}$ | $0.019_{\pm0.001}$ |
| ResNet-34 | - | $0.043_{\pm0.001}$ | $0.026_{\pm0.001}$ |
| ResNet-18 (FedWeIT) | - | $0.046_{\pm0.002}$ | $0.025_{\pm0.001}$ |
| ResNet-18 (Ours) | $n = 1$ | $0.045_{\pm0.001}$ | $0.026_{\pm0.001}$ |
| ResNet-18 (Ours) | $n = 2$ | $0.112_{\pm0.001}$ | $0.027_{\pm0.002}$ |

### F.5  Different Backbone

To further validate the effectiveness of our VFCL, we conducted experiments on datasets CIFAR100 and TinyImageNet using DenseNet (Huang et al., 2017) as the backbone. The other settings of the experiment are the same as the experiments in the main text. For experimental fairness, we try

to make sure that the number of parameters of the backbone is the same for different methods, therefore, we use DenseNet-121 in FedWeIT and Ours and DenseNet-169 in GLFC, CFeD and FedRecon. Comparisons of the number of parameters and computational overhead of the models are shown in Tab. 9.

Table 9: Comparison of different models in terms of computation complexity, including number of parameters (Params) and floating point operations (FLOPs).

|  | DenseNet-121 | DenseNet-169 | DenseNet-121 (FedWeIT) | DenseNet-121 (Ours) |
|---|---|---|---|---|
| Params | 7.98M | 14.15M | 15.88M | 15.88M |
| FLOPs | 5.76G | 6.84G | 5.82G | 5.82G |

The results of the comparative experiments are presented in Tabs. 10 and 11, where the best results are in **bold** and the second-best results are underlined. We observe that Ours outperformed the other methods by 2.0%–11.1% in terms of average accuracy, which validates the effectiveness of Ours.

Table 10: Comparison of different methods in terms of accuracy (%) on CIFAR100 with 10 tasks.

| Task | 1 | 2 | 3 | 4 | 5 | 6 | 7 | 8 | 9 | 10 | Avg. | $\Delta$ |
|---|---|---|---|---|---|---|---|---|---|---|---|---|
| FedWeIT | 81.2 | **68.3** | 60.9 | 54.7 | 50.6 | 50.2 | 49.2 | 45.9 | 43.4 | 41.5 | 54.6 | $\Downarrow$2.0 |
| GLFC | 80.5 | 65.4 | 60.0 | 51.3 | 40.3 | 41.9 | 46.1 | 45.1 | 38.8 | 39.0 | 50.8 | $\Downarrow$5.8 |
| CFeD | 80.4 | 64.9 | 56.1 | 53.6 | 48.3 | 48.0 | 45.9 | 41.9 | 42.4 | 41.2 | 52.3 | $\Downarrow$4.3 |
| FedRecon | **82.3** | 62.2 | 51.2 | 41.7 | 44.5 | 37.6 | 35.2 | 35.9 | 30.8 | 33.7 | 45.5 | $\Downarrow$11.1 |
| Ours | 76.2 | 65.2 | **62.2** | **57.1** | **55.5** | **54.9** | **51.1** | **50.3** | **47.9** | **45.2** | **56.6** | - |

Table 11: Comparison of different methods in terms of accuracy (%) on TinyImageNet with 10 tasks.

| Task | 1 | 2 | 3 | 4 | 5 | 6 | 7 | 8 | 9 | 10 | Avg. | $\Delta$ |
|---|---|---|---|---|---|---|---|---|---|---|---|---|
| FedWeIT | 80.3 | 61.8 | 56.9 | 53.4 | 52.5 | 50.3 | 48.9 | 45.8 | 44.5 | 44.0 | 53.8 | $\Downarrow$2.1 |
| GLFC | **81.6** | 59.5 | 56.9 | 55.0 | 54.0 | 50.0 | 44.5 | 44.9 | 44.8 | 43.4 | 53.5 | $\Downarrow$2.4 |
| CFeD | 80.3 | 63.4 | 59.1 | 53.6 | 51.9 | 48.7 | 50.7 | 46.0 | 42.0 | 39.7 | 53.5 | $\Downarrow$2.4 |
| FedRecon | 72.1 | 58.2 | 53.0 | 50.6 | 50.0 | 47.9 | 45.9 | 43.8 | 42.0 | 41.4 | 50.5 | $\Downarrow$5.4 |
| Ours | 74.6 | 61.5 | **59.2** | **56.4** | **54.9** | **51.8** | **54.3** | **49.4** | **48.3** | **48.6** | **55.9** | - |

## F.6 COMPARISONS OF CONVERGENCE RATES AND FORGETTING RATES

Comparisons of convergence rates are shown in Figs. 9 and 10. FedRecon could not be compared because it is a non-iterative method. The figure shows the change in accuracy from Tasks 1 to 10, where each task contained 10 rounds of global communication. It can be seen that Ours exhibits faster convergence rates as well as higher platform accuracy, and this advantage becomes more significant as the number of CL tasks increases.

Comparisons of forgetting rates are shown in Figs. 11 and 12, which compare the change in accuracy over the duration of CL for Tasks 1, 2 and 3, respectively. It can be seen that the accuracy of Ours on the learned task can remain stable in CL and is higher than the other compared methods, which validates the effectiveness of Ours in mitigating catastrophic forgetting.

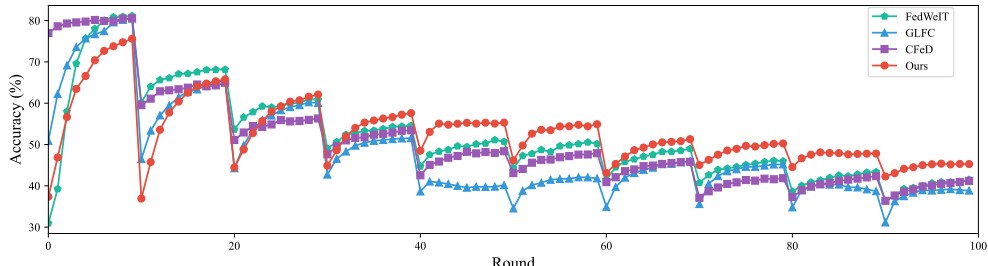

Figure 9: Comparison of convergence rates on CIFAR100.

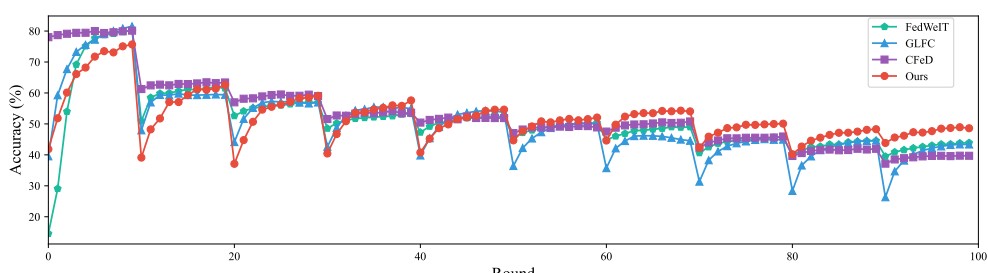

Figure 10: Comparison of convergence rates on TinyImageNet.

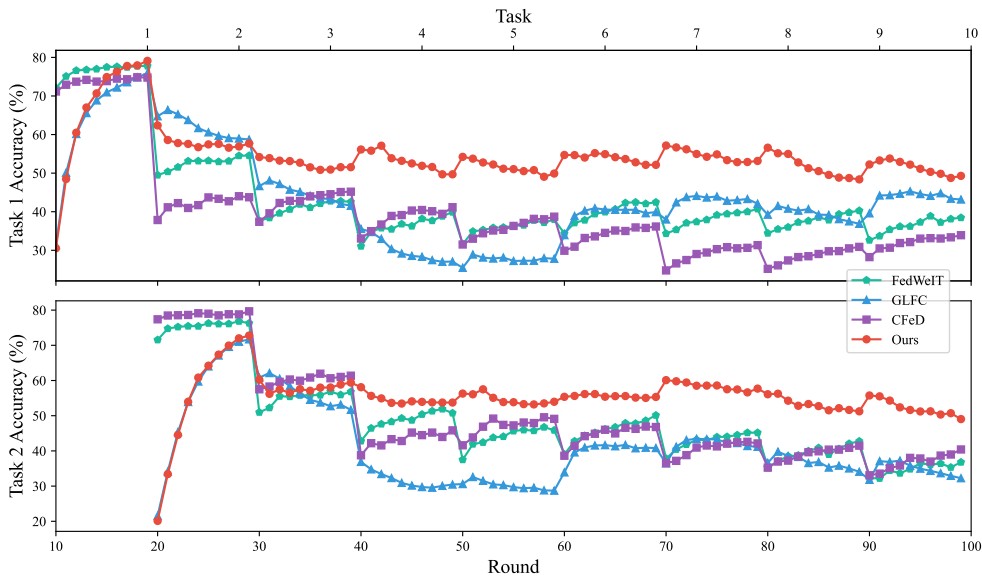

Figure 11: Comparison of forgetting rates on CIFAR100.

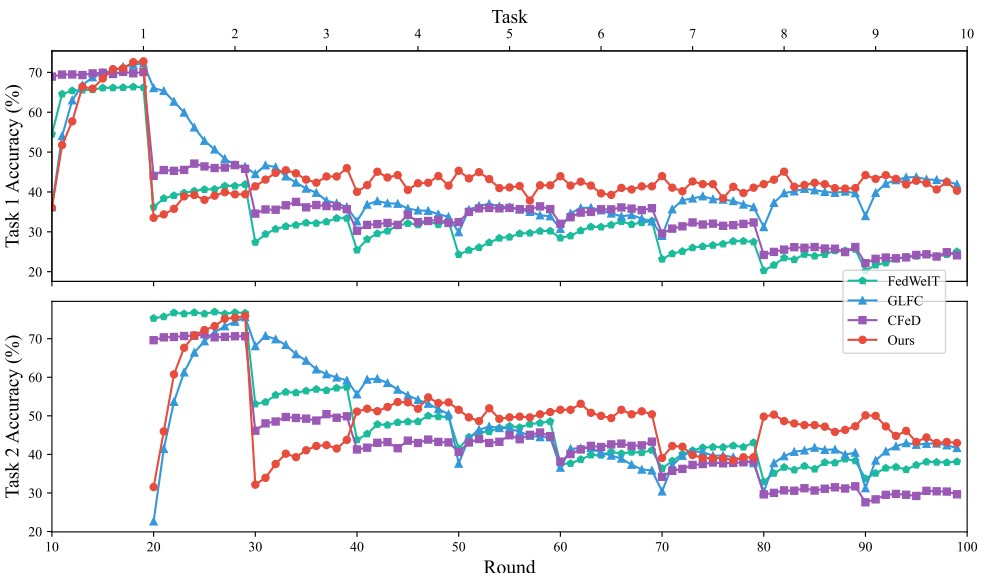

Figure 12: Comparison of forgetting rates on TinyImageNet.

