# OpenReview forum: "Variational Federated Continual Learning"
_ICLR.cc/2024/Conference — Submitted to ICLR 2024_

### Official Review · Reviewer_ouu6 · 2023-10-29

**Soundness:** 3 good
**Presentation:** 3 good
**Contribution:** 3 good
**Rating:** 6
**Confidence:** 3

**Summary:**

This paper proposes a Bayesian framework for federated continual learning. For local updates, they combine previous posterior and global posterior into a mix prior to mitigate both the catastrophic forgetting and overfitting encountered in federated continual learning. For global aggregation, they use conflation that minimizes Shannon information loss.

**Strengths:**

1. Writing is good. The problem formulation is clear.
2. They propose a comprehensive Bayesian framework to FCL, including both solutions in local updates and global aggregation.
3. They provide detailed theoretical analysis.
4. Experiments are conducted on large-scale complex datasets like CIFAR-100 and Tiny-ImageNet.

**Weaknesses:**

I'm not familiar with previous works in federated continual learning. So I cannot provide a specific assessment regarding the novelty. The performance seems pretty good so I don't have any additional feedback on the limitations of the work.

**Questions:**

How do you compare with this work?

[1] Guo, H., Greengard, P., Wang, H., Gelman, A., Kim, Y., & Xing, E. P. (2023). Federated Learning as Variational Inference: A Scalable Expectation Propagation Approach. arXiv preprint arXiv:2302.04228.

---

> ### Author Response · Authors · 2023-11-20
> **Thank you for the advice you provided !**
>
> **Comparison with "Federated Learning as Variational Inference: A Scalable Expectation Propagation Approach" (FedEP).**
>
> We compare FedEP with ours in terms of problem setting and method.
>
> Problem setting: FedEP studies the classical federated learning problem, where all clients learn a task together, where all clients jointly learn a task, and the task definition is static and predefined. In our problem setting, each client learns sequentially on a private sequence of tasks, and clients learn different tasks. Our problem is more challenging, and the challenges include catastrophic forgetting due to continual learning, as well as heterogeneity across client tasks.
>
> Method: FedEP is a centralized federated learning approach, which output is a single global model. Our approach is personalized, and it learns a personalized model for each client. The same point is that both FedEP and ours use variational inference to solve the approximate posterior.
>
> Thanks for providing this work, and we have added it to the related work.

---

> > ### Comment · Reviewer_ouu6 · 2023-11-22
> >
> > Thanks for the comparison. I would like to keep my original score.

---

### Official Review · Reviewer_J3Yt · 2023-10-30

**Soundness:** 3 good
**Presentation:** 3 good
**Contribution:** 3 good
**Rating:** 6
**Confidence:** 4

**Summary:**

The paper addresses federated continual learning (FCL). There are two novelties proposed:
1. (Primary) There have been variational generalisations, from continual learning (CL) to variational continual learning (VCL) and from federated learning (FL) to variational federated learning (VFL). This paper is the first to introduce variational federated continual learning (VFCL), by introducing variational inference with mixture prior aimed at mitigating both local overfitting and catastrophic forgetting.
2. (Secondary) In (variational) federated learning, the task of aggregating client posterior distributions into a global distribution is done via parameter average. This paper proposes an alternative: conflation.
Apart from these novelties, the paper includes theoretical analysis on generalisation bounds on VFCL's performance, and experiments and adequate ablation studies showing the advantages of the proposed approach over existing related works.

**Strengths:**

Good introduction. Good flow. Easy to read. Good review of existing works.

The contributions are clear and not difficult to understand. Many aspects of the contributions are adequately addressed with ablation studies.

The theoretical bounds in section 4 are welcome. It would be good if some connection to existing approaches can be established.

Experimental results show reasonable accuracy gains against existing approaches.

**Weaknesses:**

There are mainly 2 weak points in my view.

### Speed

The paper focuses mainly on accuracy, showing VFCL outperforming other state-of-the-art methods in a number of cases. However, the introduction of variational inference does come with complications in speed. In training, VFCL involves an additional iterative, MCMC sampling step (at optimizing eq (9)). In inference, you also need MCMC or variational inference to form the output distributions. There seems to be very little in the paper that addresses speed concerns. Whether or not speed can be a big concern for using the approach in practice seems to be left unanswered.

### Experimental Datasets

One weakness is the choice of the datasets conducted for experiments. The paper uses CIFAR100 and TinyImageNet, which are originally designed neither for federated learning nor continual learning. The non i.i.d. property was simulated. While it makes sense from a theoretical point of view, it begs the question of how the model performs in real applications. Would it still be better than existing approaches in real applications? Have you conducted your approach on any real dataset for FCL?

**Questions:**

I hope to get some feedback regarding the weak points above. Apart from those points, there are some further questions:

Eq (8) in training has to be solved using a sampling approach like MCMC. How does that impact training in terms of time?

In the experiments, Monte Carlo sampling was set to $n=1$ for the purpose of comparing with other approaches. But if the approach were to be used in a real application, is there any study to suggest in which range the value of $n$ should be?

There is an experiment on varying $\lambda_p$. Is there a similar experiment on varying $\lambda_k$? How do we know whether $\lambda_k$ is a sensitive parameter or not?

---

> ### Author Response · Authors · 2023-11-20
> **Thank you very much for your comments !**
>
> **Comparison of computational speed.**
>
> We added the comparison of training and testing time overheads of different methods in Appendix F.4. For the same number of parameters, the training and inference speeds of VFCL are almost the same as those of the other compared methods.
>
> It is worth noting that, in inference, VFCL does not perform Monte Carlo sampling, which uses the mean of the learned parameter distributions as the model parameters, and thus its inference efficiency is the same as that of a plain model.
>
>
>
> **Would our method still be better than existing approaches on real FCL datasets ?**
>
> To the best of our knowledge, existing FCL methods all utilize existing datasets to simulate FCL scenarios,  the datasets used in the comparison methods are listed in the following table:
>
> | FCL methods | Experimental Datasets                  |
> | ----------- | -------------------------------------- |
> | FedWeIT     | Overlapped CIFAR-100, NonIID-50        |
> | GLFC        | CIFAR100, ImageNetSubset, TinyImageNet |
> | CFeD        | CIFAR10, CIFAR100, Caltech-256         |
> | FedRecon    | mini-ImageNet                          |
>
> In addition, due to the privacy issues, it is very difficult to collect an FCL dataset of real scenarios, and to the best of our knowledge, there are no available FCL datasets from real scenarios, which is a research area need to be filled.
>
> To further validate the effectiveness of our method, we added the comparison experiments on Overlapped CIFAR-100 and NonIID-50, which are benchmark datasets introduced by "Federated Continual Learning with Weighted Inter-client Transfer". Detailed results are reported in Appendix F.1, and our method still outperforms the comparison method on both datasets.
>
> We consider constructing FCL datasets of real scenarios in our future work to better validate the effectiveness of the FCL methods.
>
>
>
> **How does Monte Carlo sampling affect training speed ?**
>
> We compare the training time for the number of Monte Carlo sampling $n$ from 1 to 10 in Fig. 8 of Appendix F.3. The results show that the training time is linearly correlated with $n$, the training time increases as $n$ increases. It is worth noting that $n$ does not affect the inference time, and if the training budget is sufficient, a larger $n$ can be used for better performance.
>
>
>
> **What range should be recommended for $n$ in real applications ?**
>
> We compared model accuracies for $n$ from 1 to 10 in Fig. 8 of Appendix F.3. We find that when $n$ is greater than 4, an increase in $n$ will not result in more performance gains, so we recommend setting $n$ no greater than 4.
>
>
>
> **Analytical experiments on varying $\lambda_k$.**
>
> $\lambda_k$ controls the relative weight between global and historical knowledge. We compare the accuracy and forgetting under different $\lambda_k$ in Table 7 in Appendix F.2. We found that both global and historical knowledge play important roles in model performance, and that missing either one leads to a decrease in accuracy. In addition, we found that the absence of historical knowledge causes severe forgetting.

---

> > ### Comment · Reviewer_J3Yt · 2023-11-21
> > **Thank you!**
> >
> > Thank you for your rebuttal.

---

### Official Review · Reviewer_Htsy · 2023-11-01

**Soundness:** 3 good
**Presentation:** 3 good
**Contribution:** 3 good
**Rating:** 8
**Confidence:** 5

**Summary:**

This paper looks at the problem of Federated Continual Learning, and proposes a variational inference solution where there is a variational (Gaussian) distribution over the parameters of a neural network. Previous papers have not looked at FCL using such variational methods. They use a mixture prior to mix together the priors coming from the continual learning and federated learning components. As they are interested in having different (personalised) models on each client, they have a specific global posterior aggregation method too. They also have some theoretical analysis on (upper bound of) the generalisation error of their algorithm.

**Strengths:**

1. The FCL (Federated Continual Learning) problem is interesting, and no previous methods have tried a BNN approach to it. It makes a lot of sense to try this.

2. I liked the mixture prior as a simple yet effective way to combine the terms coming from the continual learning and federated learning parts of the problem.

3. I was able to follow the method and explanations quite well (although, it should be said that I am well-versed in BNNs/VI in general). I particularly found Figure 1 helpful. (That said, a small suggestion: the plots in Figure 1 with Gaussians did not make any sense until reading the paper in detail, and mostly just distracted me.)

4. There are two experiments, and I liked the various ablations and comparisons of computation overheads.

**Weaknesses:**

1. My main issue is with the global posterior aggregation method (Section 3.3). Equation 10 makes sense to me (using a mixture of local approximate posterior distributions). However, Equation 11 does not seem related to Equation 10: Equation 11 looks like you have just simply multiplied together all the various local approximate (mean-field) Gaussians. IE it is a *product* of local posterior distributions. Am I misunderstanding something? If this is indeed the case, why write it through Equation 10 instead of just saying you are multiplying together the distributions?

2. Although it is nice to see an attempt at theoretical analysis, I did not see what benefit this brings to the paper or method. Could the authors perhaps discuss how such analysis might be helpful for current or future analysis? The three bullet points at the end of Section 4 all seemed fairly obvious to me (but perhaps that is the point? But then what use is this section?).

3. The experiments only reported one number per method. It is important to run many times (at least 3 or 5) with a mean and std deviation across runs to get an idea of if these results are significant in some way. Further questions about the experiments:
- Why are the first 2 tasks called 'warm-up tasks'?
- It is not clear exactly what is being reported in Tables 1 and 2. Is it the average accuracy after training on task t (where t is the number in the first row), or is it the final accuracy on task t's data after training on the last task? What is the Avg over? These details should ideally go in the caption (and the main text).

**Questions:**

Please see Weaknesses section.

One more minor point:
- In the second paragraph of the Introduction, the authors talk about previous methods (regularisation based and knowledge distillation based), which apparently use global knowledge / models. But then the authors say these methods do not combat local overfitting. I thought the point of use global knowledge was to combat local overfitting, so I did not understand why these other methods do not do this? I think the writing could be made clearer here.


============
Post-rebuttal

I thank the authors for the rebuttal. I have increased my score. Overall I think this paper is good and of interest to the community.

I have to say that I find using Equation 10 as part of a derivation of Equation 11 misleading, as one is a summation and the other is a product. The authors have added some clarification but I encourage them to make this more clear in order to avoid potential misunderstandings.

---

> ### Author Response · Authors · 2023-11-20
> **Thank you very much for your constructive comments !**
>
> **In global posterior aggregation, Eq. (11) does not seem related to Eq. (10).**
>
> The method described in Eq. (10) is a straightforward way to obtain a global posterior, however, it is intractable due to the non-i.i.d. client data. Although it is possible to approximate the solution using Monte Carlo sampling, the huge computational overhead and the need for a training dataset make it infeasible in practice.
>
> Therefore, we introduce the conflation, which fuses multiple distributions into a single distribution, the aggregated distribution is with density proportional to the product of the densities of input distribution, and Eq. (11) describes how the parameters in the product of Gaussian distributions are calculated, i.e., the parameters of the final aggregated distribution.
>
> We have revised this section to provide a more smooth transition and added a detailed explanation of global posterior aggregation in Appendix E.
>
>
>
> **What benefits does theoretical analysis bring to the paper ?**
>
> As a reminder, the key contribution of our approach is to integrate historical and global knowledge into the mixture prior and migrate the knowledge to the current model, mitigating local overfitting and catastrophic forgetting.
>
> This theoretical analysis proves that the mixture prior can reduce the upper bound of generalization error (as $\lambda_{p}$ increases, the overall error decreases), and clarifies the effectiveness of the mixture prior, which is the core contribution of our work.
>
> Further, we also experimentally (see Table 1) verify the positive effect of the mixed prior on the model performance, which is consistent with the findings of the theoretical analysis. In addition, we also find that too strong prior constraints, while reducing model forgetting, also reduce the model's adaptability to new tasks, ultimately undermining the model's learning ability.
>
> We have revised this section of the article to provide a clearer explanation of the utility of theoretical analysis for method.
>
>
>
> **The experiments need to be run several times to validate the effectiveness of the method.**
>
> Our comparison experiments were run 3 times. We modified tables 1,2, and added the deviation bar (due to space constraints, we only added it on "Avg.", which represents the average of the results of the rows in the table.).
>
>
>
> **Why are the first 2 tasks called 'warm-up tasks' ?**
>
> In incremental learning, when the model learns the first few tasks, the model generalization performance will be poor due to less exposure to data, and the adaptation to new tasks is slow, which is reflected in the low accuracy of the new task learning, so we call the first few tasks of incremental learning 'warm-up tasks' .
>
>
>
> **What are the evaluation metrics in Tables 1 and 2 ?**
>
> The metrics under 1-10 in the table are the average accuracies of all the tasks that have been learned, e.g. after learning the 3rd task, the accuracy under 3 is the average of the model's accuracies on tasks 1,2,3.
>
> "Avg." represents the average of the results of the rows in the table, which is the same as in "Federated Class-Incremental Learning".
>
> We revised the caption of the tables and added a detailed description of the evaluation metrics in Appendix A.
>
>
>
> **Why we say previous methods do not combat local overfitting ?**
>
> Although existing FCL works have utilized the global model as the initialization for local training, the local model will gradually diverge from the initialized global model after starting local training, and the limited local data of the client will lead to local overfitting, which reduces the generalization ability and convergence rate. Existing methods focus on solving the catastrophic forgetting without explicitly preventing the divergence of the client local learning process, so we argue that these works do not address the overfitting problem.
>
> Our method uses a personalized model on the client, and instead of replacing the local model with the weights of the global model, global knowledge is fused into the mixture prior, preventing local overfitting through regularization during the entire local training process, which can also be interpreted as passing global knowledge to the local model.
>
> We have revised the introduction to provide a clearer explanation of why previous work ignored the overfitting problem.
>
>
>
> **The confusion caused by the Gaussian notation in Fig. 1.**
>
> We revised Figure 1 to clarify the meaning of the symbols.

---

> ### Author Response · Authors · 2023-11-22
> **Kind mention on response**
>
> Dear reviewer Htsy:
>
> We greatly thank for your comments and suggestions on our submission.  We have tried our best to address all the concerns in the response, including:
>
> - Questions on global posterior aggregation.
> - Questions about the contribution of theoretical analysis to our method.
> - Suggestions and questions about the experiment.
>
> Due to the coming deadline, we sincerely hope to know your thoughts about our responses. If you have further questions, please feel free to let us know.  Thanks again.
>
> Best regards,

---

> > ### Comment · Reviewer_Htsy · 2023-12-01
> > **Response**
> >
> > I thank the authors for the rebuttal. I have increased my score. Overall I think this paper is good and of interest to the community.
> >
> > I have to say that I find using Equation 10 as part of a derivation of Equation 11 misleading, as one is a summation and the other is a product. The authors have added some clarification but I encourage them to make this more clear in order to avoid potential misunderstandings.

---

### Official Review · Reviewer_uvyG · 2023-11-03

**Soundness:** 3 good
**Presentation:** 2 fair
**Contribution:** 2 fair
**Rating:** 3
**Confidence:** 4

**Summary:**

This paper integrates the framework of variational inference with mixture model into the federated continual learning scenario.

The key idea is for each client to adopt variational inference on a mixture prior comprising its locally learned posterior & the aggregated global posterior of the previous task to optimize for a local posterior of the current task. In turn, such local posteriors will be shared with the server for aggregation (via conflation) & the aggregated posterior will be sent back to the clients so that they can update their mixture prior for the next iteration and so on.

The proposed method is compared with the latest work in FCL on two simulated continual learning scenarios on CIFAR100 & TinyImageNet.

**Strengths:**

This paper aims to address a relatively new, less addressed problem in federated learning.

The writing is mostly clear & communicates well the high-level idea.

The technical exposition is rigorously detailed, which is great. The experiments also show consistent positive results across two datasets.

**Weaknesses:**

I have the following concerns of this work (in decreasing order of importance)

First, I believe the empirical setting studied in this paper is a simplified version of the FCL setting in FedWeIT where clients have different task sequence. In this case, however, the clients are assumed to have the same task sequence, which is reducible to a pure CL setting: (1) existing CL parameterization can be adopted "as is" for the common model architecture; and (2) existing FL techniques can be applied to handle the federation of data. This ignores the essence of federated CL where there can be another cause for catastrophic forgetting which is client interference. This happens when clients have different task sequence and a naive attempt to aggregate their models (solving different tasks) can also lead to forgetting.

Thus, unfortunately, what is proposed here has not addressed that challenge because the global posterior aggregation here assumes local posteriors are derived from the same task (albeit with different data distributions).

Second, I also find the experiment too limited in comparison to the setting that was investigated under FedWeIT. For a thorough comparison, its exact same setting should have been adopted here. Also, the current reported performance is without deviation bar with relatively thin margin between best and second best methods, which is not very conclusive.

Third, I do not see how the developed theory is specific to continual and federated learning. Its theoretical setup is entirely oblivious to the FCL setting so I am not sure what it really implies here.

Last, while the writing is mostly clear, several parts still remain unclear. For example, background on conflation is missing, making it hard to see what is the loss function that replaces (10) in Section 3.3. The FL characterization in Eq. (1) is also strange: it suggests the optimization of all clients is decoupled (even thought it is not supposed to be the case). Also, are the mixing weights in Eq. (5) learnable?

**Questions:**

Unless I misunderstand this work, I believe it is focused on a setting that assumes away a key challenge of federated CL. Please let me know if I misunderstand something important here.

In addition, I'd suggest re-running the experiments on the benchmark data introduced in https://proceedings.mlr.press/v139/yoon21b/yoon21b.pdf
All experiment results should have deviation bar reported.

I also think the authors need to elaborate more on their theoretical results, and explain (if possible) how it specifically accounts for the continual and federated learning setup.

---

> ### Author Response · Authors · 2023-11-20
> **Thanks very much for your feedback and pointing out the key challenge of FCL**
>
> **Do the empirical setting ignore the catastrophic forgetting brought by client interference (a key challenge of FCL) ?**
>
> We agree that the catastrophic forgetting caused by client interference is a key challenge of FCL. However, we are afraid that the reviewer misunderstood our experimental setup. We would like to clarify that the client task sequences in our experiments are different across clients. In addition, we conducted experiments on the benchmark datasets according to the reviewer's suggestions, and the results validated the effectiveness of our method (see later answer for more details).
>
> We provide a detailed explanation of how the client private task sequences in our experiments were constructed in Appendix A. First, the original dataset is split into several tasks that contain no overlapping classes. Second, the split tasks are randomly selected as the client's task and and the classes of tasks included are also randomly selected. Private task sequences from different clients may have similar (i.e., overlapping classes), unrelated, or interfering tasks. Clients utilize the knowledge of other clients in the learning process to improve the performance of the local model while preventing interference from the knowledge of other clients.
>
> In FedWeIT, client interference is addressed through parameter decoupling and selective knowledge migration, while our approach addresses this challenge through personalization and client selection. We describe the client personalization in detail in Appendix A.2, the weights of the backbone as well as the structure of the classifiers are personalized for different clients, which allows VFCL to learn on personalized client task sequences. Client selection is described in Algorithm 1, where the server coordinates the learning of clients with similar tasks in a synchronized manner, which speeds up the convergence rate and reduces client interference.
>
> **Limited experimental setting compared with FedWeIT.**
>
> We have restated our experimental setup in the first reply, our setup is equally challenging with FedWeIT, the client task sequences are heterogeneous, capable of responding the client interference.
>
> To further validate the effectiveness of our method, we adopted the reviewer's suggestion to rerun the experiments on benchmark datasets Overlapped-CIFAR100 and NonIID-50 (repeated 3 times to obtain the mean and deviation bar), and added the detailed results to Appendix F.1, where the experimental results show that our method outperforms the others.
>
>
> **Experimental results that lack deviation bar are not conclusive.**
>
> All of our comparison experiments were repeated 3 times and averaged, and we revised Tables 1,2 to add deviation bar to the results. Due to space constraints, we only added deviation bar on "Avg.", and "Avg." denotes the average of the results in a single row of the table.
>
>
> **Relationship between theoretical analysis and method ?**
>
> In FCL, multiple tasks are learned sequentially, and each task is learned by a separate federated learning (FL) process. Since all FL processes are isomorphic, we analyze the upper bound of the generalization error of a single FL process to analyze the overall FCL process.
>
> The theoretical analysis guided our approach as follows: First, the upper bound of the generalization error of our method is affected by sample size, model capacity, and mixture prior (which is the core innovation of our work). Second, mixture prior is theoretically shown to reduce the upper bound on generalization error. Third, we obtain results in our experiments that are consistent with the theoretical analysis, verifying the positive effect of the mixture prior on the model performance.
>
>
>
> **Lack of background on conflation.**
>
> The goal of global aggregation is to find a distribution which can integrate the posterior distributions of all clients. Conflation can fuse information from multiple distributions into a single distribution, and the aggregated distribution is with density proportional to the product of the densities of input distribution. Eq. (11), which is the calculation of the parameters of the global posterior distribution, replaces the optimization process in Eq. (10).
>
> We revised this section and added a detailed description of conflation in Appendix E.
>
> **The FL characterization in Eq. (1) is strange.**
>
> The client models in our approach are personalized, so the final result obtained in Eq. (1) is all personalized models $\{\boldsymbol\theta_c\}_{c=1}^{C}$. In contrast, traditional centralized federated learning outputs only one global model, we thought this distinction might bother reviewers.
>
> **Are the mixing weights in Eq. (5) learnable ?**
>
> The mixing weights in Eq. (5) are not learnable, they can be set manually (in our paper) or calculated using existing client evaluation methods.

---

> ### Author Response · Authors · 2023-11-22
> **Kind mention on response**
>
> Dear reviewer uvyG:
>
> We sincerely appreciate your kind efforts in reviewing our paper. Now, we tried our best to clarify all the concerns in our response, including major concerns:
>
> - Concerns about the experimental setup.
> - Suggestions for re-running the experiment on the benchmark datasets introduced in FedWeIT.
> - Questions on the relationship between theoretical analysis and method.
>
> Due to the coming deadline, please feel free to let us know your thoughts, if you have further questions. Great thanks.
>
> Regards,
>
> ---

---

> ### Comment · Reviewer_uvyG · 2023-11-22
> **Thanks for the detailed update**
>
> Dear Authors,
>
> Thank you for the detailed update. I need more time to go over all the new results you have provided.
>
> For now, I have one question:
>
> In Algorithm 1 (see page 15), the server workflow will loop over each task & for each task, it selects clients who will be assigned to the task. Is it supposed to know each client's task sequence in advance?
>
> If that is the case, how will this algorithm be used in practice where the client's task sequence is not known in advance?
>
> --
>
> Furthermore, assuming that we know the task sequence in advance is the same as assuming away the client's interference.
>
> The pseudocode clearly suggests that the clients will not be trained in the order of the tasks they received. Instead, the server dictates which client solves each task at which step. This is no longer continual learning where you cannot manipulate the order in which tasks are presented at will.
>
> --
>
> Otherwise, assuming the server only selects clients that are assigned the task in the current round, doesn't it mean that it already forgot about the other clients' solution model of this task who were constructed in previous rounds?

---

> > ### Author Response · Authors · 2023-11-22
> > **Thank you very much for your reply!**
> >
> > **Does the server need to know all the task sequences for each client in advance?**
> >
> > No, the server only needs to know the task that the client is currently learning, because in practice it is not possible to determine the sequence of tasks for each client in advance. In practice, the server asks for information about the tasks of all currently available participating clients, and then selects clients that learning similar tasks for federated learning.
> >
> >
> >
> > **Does the client learn in the order of the local task sequence?**
> >
> > Yes, clients learn in the order of local continual learning. The client locally utilizes a task queue (FIFO) to manage the tasks to be learned, and the server selects clients based on the current task to be learned (i.e., the task at the head of the queue). When a client participates in training, the current task at the head of the queue is popped up, and then the head of the queue is replaced with the next task.
> >
> > The experimental setup is the same for us and several other FCL works:
> >
> > - "Federated Continual Learning with Weighted Inter-client Transfer"
> > - "Federated Class-Incremental Learning"
> > - "Continual Federated Learning Based on Knowledge Distillation"
> > - "Federated Reconnaissance: Efficient, Distributed, Class-Incremental Learning"
> >
> > and all clients learn in the order of the local task sequence.
> >
> >
> >
> > **Does the server forget what other clients have already learned?**
> >
> > No, the server maintains a global posterior at all times into which the knowledge of all clients that have participated is integrated, so the server does not forget the knowledge of tasks that have already been learned. When the client performs learning, it downloads the latest global knowledge as a prior for local training, thus migrating the existing knowledge to the local model, so it can utilize the knowledge that has already been learned.
> >
> >
> >
> > **Further explanation**
> >
> > Client selection is an optional item in our approach, and our approach can also disable client selection and deal with the challenges posed by client interference through personalization alone. The advantage of using client selection is that it reduces client interference and speeds up the convergence of individual tasks, but its disadvantage can make unselected clients wait for the next collaborative task. And not using client selection will reduce client waiting, but will bring more client task interference.
> >
> > We also verified performance without client selection in our experiments. In the experiments conducted on the benchmark datasets introduced by FedWeIT, we disabled client selection to maintain the same experimental setup as in FedWeIT, i.e., learning each client's tasks simultaneously, which are potentially similar, unrelated, or interfering. The experimental results are supplemented in Appendix F.1. The results show that we can still achieve superior performance over the comparison methods through personalization without applying client selection.
> >
> > Moreover, as a kind reminder, client selection is not the core contribution of our work, our main contribution is the integration of global and historical knowledge in a unified framework, we innovatively propose the mixture prior that stores both global and historical knowledge to simultaneously address local overfitting and catastrophic forgetting. Furthermore, to the best of our knowledge, our approach is the first BNN-based solution for FCL, providing a new technological path that is different from previous approaches.

---

### Author Response · Authors · 2023-11-20
**General Response**

We thank all the reviewers for their elaborate and constructive feedback. We made the following changes to our paper:

- We revised the second paragraph of the introduction to provide a more detailed description of overfitting.
- We revised Figure 1 to clarify the meaning of the Gaussian notation.
- We revised Section 4 to further explain the contribution of theoretical analysis to our approach.
- We added experimental details in Appendix A, including the construction flow of client private task sequences, model architecture and metrics.
- We revised Section 3.3 to provide a better explanation of global posterior aggregation.
- We provided background of conflation and a detailed explanation of global posterior aggregation in Appendix E.
- We added comparative experiments on Overlapped-CIFAR100 and NonIID-50 introduced by FedWeIT in Appendix F.1.
- We added analytical experiments on $\lambda_{k}$  in Appendix F.2.
- We added analytical experiments on the number of Monte Carlo sampling $n$ in Appendix F.3.
- We compared the training and testing time overheads of the different methods in Appendix F.4.

---

### Meta-Review · Area_Chair_Q8Ai · 2023-12-06

**Metareview:**

The manuscript presents a federated continual learning based on variational inference, complementing local and global posteriors to avoid overfitting to individual clients, while learning a sequence of tasks of individual clients.

Strength:
1. The manuscript addresses a relatively new, less addressed problem in federated learning  using variational inference (Reviewer uvyG, Htsy, J3Yt and ouu6)
2. Authors provide a detailed theoretical analysis (Reviewer uvyG, Htsy, and ouu6)
3. PErformance studies are satisfactory (All reviewers)

Weakness:
1. The server defines a sequence of the task - and this reduces the premise of the proposed federated learning problem, and the server is still privy to the tasks in each client.. There are several methods in literature to address this reduced problem, and manuscript doesn't compare against these existing methods too  (Reviewer uvyG)
2. I have to say that I find using Equation 10 as part of a derivation of Equation 11 misleading, as one is a summation and the other is a product. The authors have added some clarification but I encourage them to make this more clear in order to avoid potential misunderstandings.(Reviewer Htsy)

**Justification For Why Not Higher Score:**

The manuscript addresses a very novel problem and uses a clever trick using variational inference to handle this (aligned with the past in ICLR on VCL, UCB etc.). However, there are discrepancies in reviewer feedbacks and would need clarification on this front.

**Justification For Why Not Lower Score:**

N/A

---

### Decision · Program_Chairs · 2024-01-16

Reject